# Absence of a dissipative quantum phase transition in Josephson junctions: Theory

BY CARLES ALTIMIRAS, DANIEL ESTEVE, ÇAĞLAR GIRIT, HÉLÈNE LE SUEUR, PHILIPPE JOYEZ[a]

Université Paris-Saclay, CEA, CNRS, SPEC
91191 Gif-sur-Yvette Cedex, France

[a]. philippe.joyez@cea.fr

*Date: October 30, 2024*

### Abstract

We investigate the resistively shunted Josephson junction (RSJ) at equilibrium, using exact path integral techniques. Our writing of the effective action makes it clear that the superconducting-insulating quantum phase transition long believed to occur in the RSJ, cannot exist. This can be traced to translational invariances in the Caldeira-Leggett Hamiltonian making the junction's reduced ground state highly degenerate and actually lacking the symmetry that the transition is supposed to break. For all parameters, we find that shunting a junction makes it more superconducting. We reveal that the UV cutoff of the resistor plays an unforeseen key role in these systems, and show that the erroneous prediction of an insulating state resulted in part from ill-assuming it would not. Our results fully support and confirm the experimental invalidation of this quantum phase transition by Murani *et al.* in 2020.

## Table of contents

# 1  Introduction

In the early 1980's Caldeira and Leggett [1] introduced a Hamiltonian allowing a rigorous quantum-mechanical description of dissipation in circuits connected to a Josephson junction (JJ). Using this Hamiltonian, they predicted quantitatively how dissipation reduces quantum tunneling of the junction's phase –a macroscopic electrical variable– and it was precisely confirmed experimentally a few years later [2].

Shortly after Caldeira and Leggett introduced their modeling of dissipative systems, Schmid [3] predicted that a dissipative quantum phase transition (QPT) should occur for a quantum particle in a 1D periodic potential submitted to friction : Above a well-defined threshold in the friction strength, independent of the potential depth and particle mass, the particle localizes in one well of the potential, while below this threshold it is delocalized, in apparent continuity with the Bloch states that exist in absence of friction.

At the end of his Letter [3], Schmid briefly mentions a resistively shunted Josephson junction (RSJ) is analogous to the system he considers and suggests one could use it as a test bed to observe his predicted *localization* effect. In this analogy, the phase of the junction plays the role of the particle's position, the friction strength scales as $R^{-1}$, the inverse of the shunt resistance, and Schmid's analogy implies the junction's phase should be localized only when the shunt resistance $R$ is smaller than $R_Q = h/4\,e^2 \simeq 6.5\,\mathrm{k\Omega}$ and delocalized when $R > R_Q$, irrespective of the junction's characteristics (size, transparency, material . . . ). The standard interpretation of this localization|delocalization dissipative QPT is that at $T = 0$, the JJ should be superconducting for resistances $R < R_Q$ and *insulating* for $R > R_Q$. Even though this predicted insulating phase strangely conflicts with the perturbative limit $R \to \infty$ and the classical understanding of JJs (see Appendix A), theoretical papers that examined the subject using many different techniques have, to the best of our knowledge, all essentially confirmed this interpretation [4, 5, 6, 7, 8, 9, 10, 11, 12, 13, 14, 15] and the phenomenon was linked with quantum impurity problems [16].

In 2020 Murani *et al.* [17] (including most of the present authors) used state-of-the-art experimental techniques to investigate squids shunted with resistances $R \geqslant 1.2\,R_Q$ and observed no sign of a quantum critical behavior [18]: a dc magnetic flux was modulating the measurements (implying the squid loop hosted a dc supercurrent, hence not being insulating), with no $T$-power-law dependence of the modulation amplitude at low temperatures. Based on these experimental observations, Murani *et al.* concluded to the absence of the insulating state predicted by the standard interpretation of Schmid's analogy for Josephson junctions. Subsequently, a few papers [19, 20, 21, 22, 23, 24, 25] explicitly reaffirmed the existence of Schmid's "insulating state" in Josephson junctions, at least in some parameter domain. Thus, the scientific community has not yet attained a consensus regarding the presence or absence of Schmid's QPT in JJs. This underscores the current lack of a comprehensive theoretical understanding of the RSJ.

In this work, we bring theoretical support to the conclusion of Murani *et al.* that RSJs are always superconducting in their ground state. To do so, we start from the Caldeira-Leggett description of a junction connected to an arbitrary linear admittance. We then use an exact method based on the path integral formalism for obtaining the equilibrium reduced density matrix (RDM) of the junction, together with key observables characterizing the junction's transport properties. Already at the qualitative level of the equations, we give an argument ruling out the existence of the predicted QPT. Our numerical results show that, for all parameters tested, a resistively shunted Josephson junction is always more superconducting than the same unshunted junction. By highlighting differences between works predicting the QPT on one side and our present work and experiments on the other side, we elucidate how they came to erroneously predict a quantum phase transition.

## 2  Description of the system

We model the effect of dissipation on a Josephson junction in the same way as Caldeira and Leggett (CL) [1], with a bath of LC harmonic oscillators providing a linear viscous damping force proportional to the voltage across the junction (*i.e.* the time derivative of the junction's phase), independently of the value of the phase. The corresponding Hamiltonian is

$$H = E_C\,N^2 - E_J\cos\varphi + \sum_n 4\,e^2\,\frac{N_n^2}{2\,C_n} + \varphi_0^2\,\frac{(\varphi_n - \varphi)^2}{2\,L_n},\tag{1}$$

where $\varphi_0 = \hbar/2\,e$ is the reduced flux quantum, $\varphi$ (resp. $N$) denotes the junction's phase (resp. number of Cooper pairs on the capacitor) which are conjugate and verify $[\varphi, N] = i$, and the $\varphi_n$ (resp. $N_n$) denote the phase (resp. dimensionless charge) of the bath harmonic oscillators.

The bath oscillators are in infinite number, forming a continuum in the frequency domain, characterized by the spectral density of modes

$$J(\omega) = \frac{\pi}{2}\sum_{n=1}^N \omega_n^2\,Y_n\,\delta\,(\omega - \omega_n) = \omega\,\mathrm{Re}Y(\omega),$$

where $\omega_n = 1/\sqrt{L_n\,C_n}$ is the $n^{\text{th}}$ mode angular frequency, $Y_n = \sqrt{C_n/L_n}$ its admittance, and $Y(\omega)$ the admittance formed by the continuum. Although this model and the numerical technique we employ below can handle any form of the admittance, we will focus here on the so-called Ohmic case where $\mathrm{Re}Y\,(\omega = 0) = 1/R$, with $R$ the dc shunting resistance, such that $J(\omega)$ is linear in frequency at low frequency. For fundamental reasons, any concrete dissipative bath has a UV cutoff frequency [1]. Here, we assume that $\mathrm{Re}Y(\omega)$ has a Lorentzian shape

$$\mathrm{Re}Y(\omega) = \frac{R^{-1}}{1 + (\omega/\omega_c)^2}\tag{2}$$

which would correspond to a $L\,R$ series circuit, with $\omega_c = R/L$. In a practical implementation of a metallic resistor, the inductance $L$ would be at least the geometrical inductance of the device. Our approach nevertheless allows considering the theoretical $\omega_c \to \infty$ limit.

The quadratic forms where the junction's phase appears in the last term of (1) can be expanded, giving

$$H = H_{\mathrm{CPB}} + H_{\mathrm{bath}} + H_{\mathrm{coupling}} + H_{\mathrm{CT}}$$

with the different parts corresponding respectively to a bare Cooper pair box (CPB)

$$H_{\mathrm{CPB}} = E_C\,N^2 - E_J\cos\varphi,$$

the uncoupled bath of harmonic oscillators

$$H_{\mathrm{bath}} = \sum_n \frac{(2\,e\,N_n)^2}{2\,C_n} + \frac{(\varphi_0\,\varphi_n)^2}{2\,L_n} = \sum_n \hbar\,\omega_n\left(a_n^+\,a_n + \frac{1}{2}\right),$$

the coupling term

$$H_{\mathrm{coupling}} = -\varphi_0\,\varphi \times \left(\sum_n \varphi_0\,\frac{\varphi_n}{L_n}\right) = -\varphi_0\,\varphi \times I_Y$$

where the junction phase $\varphi$ couples to the current $I_Y$ flowing in the admittance $Y(\omega)$, and the so-called counter-term

$$\begin{aligned}
H_{\mathrm{CT}} &= (\varphi_0\,\varphi)^2 \sum_n \frac{1}{2\,L_n} \\
&= (\varphi_0\,\varphi)^2 \int_0^\infty \frac{\mathrm{d}\,\omega}{\pi}\,\mathrm{Re}Y(\omega) = \frac{(\varphi_0\,\varphi)^2}{2\,L} \\
&= E_L\,\varphi^2,
\end{aligned}$$

which appears as a parabolic inductive potential term for the junction phase and which is essential for having the expected damped equations of motion in the classical limit [1]. Interestingly, the counter-term transforms our CPB Hamiltonian into a fluxonium [26] Hamiltonian at zero external flux

$$H_{\mathrm{CPB}} + H_{\mathrm{CT}} = H_{\mathrm{Fl}}.$$

At this point we highlight that the coupling term scales as $1/R$, making it perturbative in the large $R$ limit. Note that if one considers a fixed cutoff frequency, the counter-term inductive energy $E_L$ also vanishes as $1/R$ (since $1/L = \omega_c/R$). Thus, in this Caldeira-Leggett model, a very large shunt resistor appears as a perturbation to the CPB, in agreement with the intuitive expectation that when $R$ increases to infinity no current can flow into it, so that dissipation disappears and one can just remove the resistance from the circuit. In the case of a purely inductive shunt with $L \to \infty$, one also recovers the physics of a CPB [27] (but we will not appeal to this result in the following).

## 3 Equilibrium reduced density matrix from path integrals

The equilibrium reduced density matrix (RDM) of the CPB (i.e. the junction and its capacitor) at temperature $T$ is obtained as

$$\rho_\beta = \frac{1}{Z}\mathrm{Tr}_{\mathrm{b}}\,e^{-\beta H}\,. \tag{3}$$

where $\beta = (k_B T)^{-1}$ is the inverse temperature, $Z = \mathrm{Tr}[\exp(-\beta H)]$ is the partition function of the entire system, and $\mathrm{Tr}_b$ corresponds to tracing out the bath oscillators. For the linear coupling term and the harmonic bath we have, this tracing out can be performed exactly, yielding the matrix elements of the RDM in coordinate representation as a path integral in imaginary time [1, 28, 29, 30]

$$\rho_\beta[\phi, \phi'] = \frac{1}{Z}\int \mathcal{D}\varphi \exp\left[-\frac{1}{\hbar}\left(S_{\mathrm{Fl}}^E[\varphi] + \Phi[\varphi]\right)\right], \tag{4}$$

where the functional integral is over all imaginary time paths $\varphi(\tau)$ having the boundaries $\varphi(0) = \phi$ and $\varphi(\hbar\beta) = \phi'$. In this expression, the terms in the exponential respectively denote the Euclidean action of the fluxonium

$$S_{\mathrm{Fl}}^E[\varphi] = \int_0^{\hbar\beta} d\tau\,\mathcal{L}_{\mathrm{Fl}}[\varphi], \tag{5}$$

with $\mathcal{L}_{\mathrm{Fl}}[\varphi] = \frac{\hbar^2}{4\,E_C}\,\dot\varphi^2 - E_J \cos\varphi + E_L\,\varphi^2$, the Lagrangian of the fluxonium and

$$\Phi[\varphi] = -\frac{1}{2}\int_0^{\hbar\beta} d\tau \int_0^{\hbar\beta} d\tau'\, \varphi(\tau)\, K\,(\tau - \tau')\,\varphi(\tau')\;, \tag{6}$$

the Feynman-Vernon influence functional [28], with the kernel

$$K(\tau) = \frac{R_Q}{2\,\pi}\, S_{II}\,(-i\,\tau) \tag{7}$$

where $S_{II}$ is the equilibrium autocorrelation function of the current in the admittance (shunted at its ends). For $t \in \mathbb{R}$, $S_{II}(t)$ is obtained using the quantum fluctuation-dissipation theorem and the Wiener-Khinchin theorem

$$S_{II}(t) = 2\int_{-\infty}^{\infty} \hbar\,\omega\, \mathrm{Re}Y(\omega) \frac{e^{-it\omega}}{(1 - e^{-\beta\hbar\omega})}\,\frac{d\,\omega}{2\,\pi} \tag{8}$$

which shows that without a UV cutoff in $\mathrm{Re}Y(\omega)$, $S_{II}\,(t \in \mathbb{R})$ would be divergent and hence nonphysical. In Eq. (7) this expression is simply prolonged to complex times, yielding

$$K(\tau) = \frac{R_Q}{2\,\pi}\int_0^{+\infty} \hbar\,\omega\, \mathrm{Re}Y(\omega) \frac{2\cosh\left[\left(\frac{\beta\hbar}{2} - \tau\right)\omega\right]}{\sinh\frac{\beta\hbar\omega}{2}}\,\frac{d\,\omega}{2\,\pi} \tag{9}$$

and one can check that $\int_0^{\beta\hbar} d\tau\, K(\tau) = 2\,E_L$. In Appendix B, we provide analytical expressions for $K(\tau)$, for the Lorentzian $\mathrm{Re}Y(\omega)$ we consider. In Appendix D, we show that Schmid and other authors use an effective action equivalent to ours (although the counter-term is not explicitly present in their writing), with the kernel taken in the $\omega_c \to \infty$ limit (and Schmid furthermore considers the $T \to 0$ limit).

## 3.1 Hubbard-Stratonovich transformation

For evaluating the path integral (4), we then rewrite the influence functional by means of a Hubbard-Stratonovich transformation. In this process, one introduces an auxiliary random scalar field $\xi(\tau)$ having Gaussian fluctuations verifying

$$\langle \xi(\tau)\,\xi(\tau')\rangle = S_{II}\,(-i\,(\tau - \tau')), \tag{10}$$

such that the double integral in Eq. (6) involving $\varphi$ at two different imaginary times can be replaced by a single integral involving $\varphi$ at only one time, averaged over all possible realizations of $\xi$ [31, 32, 33]. Upon this exact transformation, Eq. (4) becomes

$$\rho_\beta\,[\phi, \phi'] = \frac{1}{Z}\int \mathcal{D}\xi\; W[\xi]\int \mathcal{D}\varphi \exp\left[-S_{\mathrm{Fl}}^E[\varphi] - \frac{1}{\hbar}\int_0^{\hbar\beta} d\tau\, \xi(\tau)\,\varphi_0\,\varphi\right]$$

with a Gaussian weight functional $W[\xi]$ ensuring Eq.(10). In the last expression, the terms in the exponential can be seen as the Euclidean action of a fictitious system made of a fluxonium coupled to a given realization of a random "current noise" $\xi(\tau)$ due to the bath, so that Eq. (4) is now reformulated as

$$\rho_\beta\,[\phi, \phi'] = \frac{1}{Z}\int \mathcal{D}\xi\; W[\xi]\int \mathcal{D}\varphi \exp\left[-\frac{1}{\hbar}S_{\mathrm{Fict}}^E\,[\varphi, \xi]\right], \tag{11}$$

with

$$S_{\text{Fict}}^{E}[\varphi, \xi] = \int_0^{\hbar\beta} d\tau \left( \frac{\hbar^2}{4\,E_C}\,\dot{\varphi}^2 - E_J \cos\varphi + E_L\,\varphi^2 + \xi\,\varphi_0\,\varphi \right). \tag{12}$$

### 3.1.1  Invalidation of Schmid's QPT

At this point, one can realize that valid states for these equations all have a finite extent in $\varphi$. Indeed, in the action of Eq. (12), when $0 < E_L = \hbar\omega_c\,R_Q/4\pi R$, the counter-term $E_L\,\varphi^2$ acts as a confining potential since it dominates other terms at large $|\varphi|$ (the random noise $\xi$ is $\varphi$-independent, and Gaussian-distributed with a finite variance $S_0$ for its mean value given by (19)). Hence, as long as $\omega_c > 0$ and $R < \infty$, the ground state of our equations is localized for all parameters. Given the link between Schmid's action and ours (see Appendix D), the localized ground state yielded by our equations is also a valid ground state for his action, for any $R < \infty$. This rules out the ground state localization|delocalization transition Schmid predicts at $R = R_Q$.

The always-localized states we obtain may seem to break the discrete translational symmetry present in the Caldeira-Leggett Hamiltonian (1) and to be in conflict with the intuitive understanding of Schmid's prediction presented in the Introduction that, at weak damping, the states of the system should resemble the Bloch states that exist in absence of damping. In Appendix E, we show that there is actually no problem there: the translational symmetry of the Hamiltonian makes the states infinitely degenerate in that system, such that, for $R < \infty$, one can exhibit infinitely many ground states, either localized (like ours) or non-dispersing Bloch-like delocalized with respect to the junction's phase. However, since the junction's phase is not an observable, all these ground states are indiscernible, which makes discussing about a localization|delocalization QPT futile.

Although we could conclude now on these qualitative arguments, in the following we show that our approach enables for the first time to make quantitative numerical predictions for the RSJ for all possible parameters. We illustrate this for various parameters, with some qualitative understanding of the observed variations. In the process, we explain why previous authors came to predict a QPT.

## 3.2  Stochastic Liouville equations

For any given realization of $\xi(\tau)$ in Eq. (11), the integral of the action of the fictitious system over all $\varphi$ paths can be seen as an element $\rho_\xi\,[\phi, \phi']$ of a (non-normalized) RDM obeying the imaginary-time stochastic Liouville equation

$$-\hbar\frac{\partial}{\partial\tau}\,\rho_\xi = (H_{\text{Fl}} + \xi(\tau)\,\varphi_0\,\varphi)\,\rho_\xi \tag{13}$$

of the fictitious fluxonium coupled to the noise source $\xi(\tau)$, so that (11) reads

$$\rho_\beta\,[\phi, \phi'] = \frac{1}{Z}\int \mathcal{D}\xi\ W[\xi]\,\rho_\xi\,[\phi, \phi'].$$

The later equation translates into a path integral equation for the RDM operators, independently of any choice of basis

$$\rho_\beta = \frac{1}{Z}\int \mathcal{D}\xi\ W[\xi]\,\rho_\xi. \tag{14}$$

For obtaining the physical equilibrium RDM of the CPB one then needs to perform the remaining path integral over $\xi$ in Eq. (14). This can be done using the following scheme. For a given realization of $\xi(\tau)$, one starts with $\rho_\xi\,(\tau = 0)$ an equipartitioned diagonal matrix (corresponding

to an infinite temperature state of the fictitious fluxonium, appropriate for $\tau = 0$) and integrates (13) up to $\rho_\xi (\tau = \hbar \beta)$. This yields a non-normalized RDM matrix with no particular physical meaning. Repeating this numerical integration for a suitable number of drawings of the random noise obeying Eq. (10) amounts to sampling $W[\xi]$, and the normalized average of the different $\rho_\xi (\hbar \beta)$ is expected to converge to the physical equilibrium RDM

$$\frac{\sum \rho_\xi (\hbar \beta)}{\mathrm{Tr} \sum \rho_\xi (\hbar \beta)} \to \rho_\beta.$$

We stress that if this stochastic averaging converges properly, the resulting density matrix is exact; it takes into account the interaction of the system and the bath to all orders with no approximation. Let us also note that the above path integral method can be applied to any open system at equilibrium where position-like degrees of freedom are linearly coupled to a linear bath. It can be extended to cases where the system-bath coupling is a non-linear function of the system's coordinates [33]. It can even be extended to real-time out-of-equilibrium dynamics of the system [31, 33] at the price of introducing additional complex cross-correlated real-time stochastic variables.

### 3.2.1 Numerical implementation

For the numerical implementation of the above stochastic method, we choose as working basis the $\mathcal{K}$ lowest eigenstates $\{|\Psi_k\rangle, 0 \leqslant k \leqslant \mathcal{K} - 1\}$ of the uncoupled fluxonium (the expected finite extent of the ground state in $\varphi$ ensures that such truncation is possible). For obtaining these eigenstates, we use an intermediate discretized phase basis $\{\varphi_j = j \, \delta \varphi, \delta \varphi \ll 2 \, \pi, j \in \mathbb{Z}, |j| < \varphi_{\max} / \delta \varphi\}$, with $N^2 = -\partial^2 / \partial \varphi^2$ approximated as a finite difference, so that the Hamiltonian is a tridiagonal matrix in this discretized phase basis. Optimized diagonalization routines yield the first a few hundred eigenstates of such tridiagonal matrices very fast, even when $\pm \varphi_{\max}$ spans many wells of the cosine (low $E_L$).

Note that our working basis is *very* different from that of the bare CPB which is the reference system we are interested in; this fluxonium basis has notably a much greater density of levels [27]. At low temperature, the most relevant energy scale for the bare CPB is its transition energy from the ground state to the first exited state $\hbar \omega_{01} = E_1 - E_0$ at zero offset charge (see Appendix C), which varies from $\hbar \omega_{01} \simeq E_C$ when $E_C \gg E_J$ to $\hbar \omega_{01} \simeq \sqrt{E_J E_C}$ in the opposite limit $E_J \gg E_C$. This is the "natural" energy scale we consider in the following, not the transition frequencies of the fluxonium. We choose the truncation of the working basis to encompass all the energy scales we take into account (and $\varphi_{\max}$ in the intermediate basis is set accordingly).

Then, in the working basis, the stochastic differential Liouville equation (13) is numerically integrated using discrete imaginary times steps $\{\tau_m = m \, \delta \tau\}$, with $\delta \tau = \hbar \beta / M$ and $0 \leqslant m \leqslant M - 1$, and starting with $\rho (\tau = 0) = I_\mathcal{K} / \mathcal{K}$, with $I_\mathcal{K}$ the identity matrix. The actual approximate integration of (13) is performed using the symmetric Trotter iteration scheme

$$\rho_\xi(\tau_{m+1}) = \exp \left( \varphi_0 \, \varphi \, \xi(\tau_m) \frac{\delta \tau}{2 \, \hbar} \right) \cdot \exp \left( -H_{\mathrm{Fl}} \frac{\delta \tau}{\hbar} \right) \cdot \exp \left( \varphi_0 \, \varphi \, \xi(\tau_m) \frac{\delta \tau}{2 \, \hbar} \right) \cdot \rho_\xi(\tau_m) \qquad (15)$$

that preserves the positivity of the RDM at each step [34]. In Appendix B, we explain how we generate the random noises $\xi(\tau_m) \, \delta \tau$ entering this iteration scheme.

As explained above, after numerically integrating Eq. (13) for $P$ different realizations of $\xi$, we take the average RDM as

$$\bar{\rho} = \frac{\sum_{p=1}^P \rho_{\xi_p} (\hbar \beta)}{\mathrm{Tr} \sum_{p=1}^P \rho_{\xi_p} (\hbar \beta)}. \qquad (16)$$

In the large $P$ limit this averaged RDM is expected to tend to the true equilibrium RDM, which must be Hermitian and positive-semidefinite. After a finite number of drawings, $\bar{\rho}$ is not perfectly Hermitian-symmetric, however, it is legitimate to symmetrize it. Indeed, for the problem we consider and in the basis we use, for a given drawing of the $\{\xi(\tau_m)\}$ yielding $\rho_\xi$, drawing the reversed sequence $\{\xi(\tau_{M-1-m})\}$ is equally probable and would yield the transposed of $\rho_\xi$ (in our working basis, all the matrices in (15) are real). Hence, for each drawing we may just add $\rho_\xi$ and its transposed matrix to our stochastic average, so that it always remain (Hermitian-) symmetric and positive-semidefinite (up to numerical accuracy). Note that even without such symmetrization, when the average converges properly (see below), the asymmetry of $\bar{\rho}$ reduces as $P$ increases, such that symmetrizing or not the RDM does not perceptibly change the expectation values of the observables we consider below.

While obtaining the RDM we can simultaneously evaluate expectation values of any operator $A$, as

$$\langle A \rangle = \operatorname{Tr} \bar{\rho}\, A \;\; = \;\; \frac{\sum w_p \operatorname{Tr} \hat{\rho}_p\, A}{\sum w_p} = \frac{\sum w_p\, a_p}{\sum w_p}$$

where $w_p = \operatorname{Tr} \rho_{\xi_p}(\hbar\beta)$, $\hat{\rho}_p = \rho_{\xi_p}(\hbar\beta)/w_p$ is the normalized RDM resulting from the integration of Eq. (13) with the $p^{\text{th}}$ noise realization and $a_p = \operatorname{Tr} \hat{\rho}_p\, A$ the corresponding (nonphysical) expectation value of $A$. In this expression, the trace of the $\rho_{\xi_p}(\hbar\beta)$ hence appear as the weight of a given noise realization in the final estimate of any expectation value (drawings with large traces correspond to paths with lower action in the path integral). The error bars on the estimated expectation value are obtained from the estimator of the variance of the weighted average using the Central Limit Theorem and the effective number of data points $P_{\text{eff}}(P) = (\sum w_p)^2 / \sum w_p^2$.

At large shunt resistance values and high temperature, the $\{w_p\}$ are such that the effective number of samples $P_{\text{eff}}(P)$ grows fast with the number of drawings $P$ and the weighted means converge well. However, when reducing $R$ (i.e. increasing the coupling to the bath) at fixed $E_C, E_J, \hbar\omega_c$ and $kT$, one must increase the number of time steps needed to keep the random increments $\xi(\tau_m)\, d\tau$ small enough, but after the random walk integration of Eq. (13) this nevertheless translates into an increased variance of the $\{w_p\}$, and a corresponding reduction of $P_{\text{eff}}$. At some point in this increase, the weighted estimation of the expectation values becomes dominated by the few drawings that fall in the (positive side) tail of the $w_p$ distribution. In other words, when $R$ is much reduced, the average is dominated by very few drawings (and possibly a single one when $P_{\text{eff}} \simeq 1$ and no longer grows substantially with $P$). This indicates that, in this case, the action has a deep and sharp minimum representing only an extremely small volume in the phase space of the $\xi$ noise, making the method extremely inefficient. In such case, whether or not the method can still yield reliable estimates of observables depends on the derivatives of those observables around this minimum. Similarly, when reducing the temperature (all other parameters kept fixed), the number $M$ of steps in $\tau$ also needs to be scaled up, eventually causing the same poor statistics. The $R$ and $T$ ranges where the statistics are poor depend on the other system parameters, and notably on the cutoff frequency. The data presented below are all in regimes where the estimators have small error bars, away from these problematic limits.

## 4  Results

In Fig. 1 we show the expectation values of the rms charge fluctuations $\sigma_N = \langle N^2 \rangle^{1/2}$ and the effective Josephson coupling $\langle \cos\varphi \rangle$ as a function of the reduced temperature $kT/\hbar\omega_{01}$, for different values of $E_J/E_C$, for the RSJ at large values of $R/R_Q$. The finite values reached by these expectation values at low temperature attest that the junction allows (super)current flow in its ground state. Indeed, if the junction were insulating, its effective inductance $L_{\text{eff}} = \varphi_0^2/E_J\langle\cos\varphi\rangle$ would be infinite (the Josephson coupling $E_J\langle\cos\varphi\rangle$ vanishes) and the charge $N$ on the capacitor would fluctuate just as in the $C\|Y$ circuit, yielding

$$\sigma_{N,C\|Y} = \frac{C}{2e} \left( \int_{-\infty}^{\infty} \hbar\omega \operatorname{Re} \frac{1}{iC\omega + Y(\omega)} \coth\left(\frac{\hbar\omega}{2kT}\right) \frac{d\omega}{2\pi} \right)^{1/2}.$$

For the Lorentzian admittance (2), this reaches the zero point fluctuations

$$\sigma_{N,C\|Y}(T=0) = \left( \frac{\hbar\,\omega_c}{4\,\pi\,E_C} \frac{\log\left(\sqrt{\alpha}+\sqrt{\alpha-1}\right)}{\sqrt{\alpha}\,\sqrt{\alpha-1}} \right)^{1/2}, \tag{17}$$

with $\alpha = \frac{\pi R \hbar \omega_c}{4 R_Q E_C}$. The conducting character of the JJ is evidenced by the fact that $\sigma_N$ saturates to values larger than $\sigma_{N,C\|Y}(T=0)$, consistently with the finite saturation value of $\langle \cos\varphi \rangle$.

In that Figure, we also compare our numerical results for these observables to those obtained for the thermal averages of the bare CPB considering all gate charge values (see Appendix C). For these large resistances, most of the numerical expectation values for the RSJ are found close to that of the CPB. At low temperatures they are found slightly above those of the CPB, but by increasing further the resistance (data not shown) one recovers more closely the bare CPB results, as expected for a vanishing perturbation. At large temperatures, some results for $\sigma_N$ are slightly below the asymptote $\sqrt{kT/2\,E_C}$ (valid for both the bare CPB and the $C\|Y$ circuit), which we attribute to our basis truncation.

In Fig. 2 we consider the $R$-dependence of the same expectation values for different ratios $E_J/E_C$ and at the low temperature $kT = 0.01\,\hbar\omega_{01}$. We observe that both $\langle \cos\varphi \rangle$ and $\sigma_N$ smoothly increase when $R$ is reduced. In Fig. 3, we show that for large resistance values, large $E_C/E_J$ and at the low temperature $kT = 0.005\,\hbar\omega_{01}$, changing the cutoff frequency $\omega_c$ of the environment admittance has a weak effect at small $\omega_c$, while at large $\omega_c$, the expectation values of the RSJ do depend on the actual value of the bath cutoff, the junction becoming more superconducting as $\omega_c$ increases. Similar behavior is observed for other $E_C/E_J$ ratios and shunt resistances values.

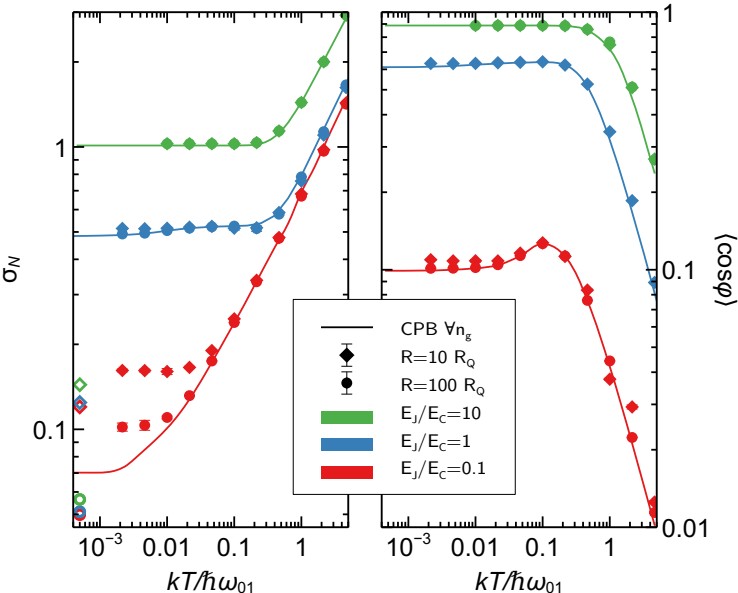

**Figure 1.** Temperature dependence of the rms charge fluctuations $\sigma_N$ on the capacitor (left panel) and the Josephson coherence factor $\langle \cos\varphi \rangle$ (right panel) for large shunt resistance values and different $E_J/E_C$ ratios, for $\hbar\omega_c = 0.4\,\hbar\omega_{01}$. In both panels, the solid lines are the thermal expectation values for the unshunted CPB allowing any gate charge (See Appendix C). For larger resistance values, the calculated expectation values (markers) are getting closer to the bare CPB values, as expected for a vanishing perturbation. Open symbols in the left panel are the zero temperature limits of $\sigma_{N,C\|Y}$ (Eq. (17)) with the same $Y(\omega)$ (same resistance and cutoff) as the filled symbol of corresponding color and shape. The fact that $\sigma_N$ saturates above these values shows that the junction has finite supercurrent fluctuations in its ground state, consistent with the finite value of the Josephson coherence in the right panel.

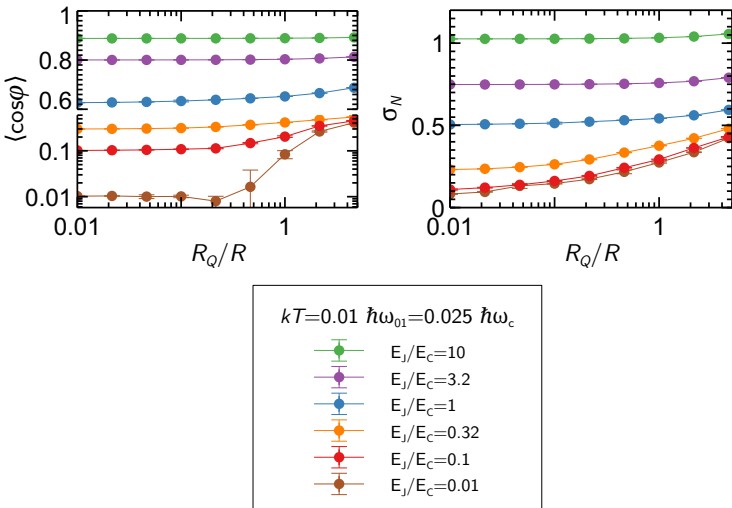

**Figure 2.** Resistance dependence of the Josephson coherence factor $\langle\cos\varphi\rangle$ (top left panel - note the log-lin broken vertical axis) and charge fluctuations on the capacitor (top right panel) for different $E_J/E_C$ ratios at the low temperature $kT = 0.01\,\hbar\omega_{01}$ and for $\hbar\omega_c = 0.4\,\hbar\omega_{01}$. One observes that both $\langle\cos\varphi\rangle$ and $\sigma_N$ increase when reducing the value of the shunt resistance and tend to saturate at the bare CPB value at large $R$. No change of behavior is observed around $R = R_Q$.

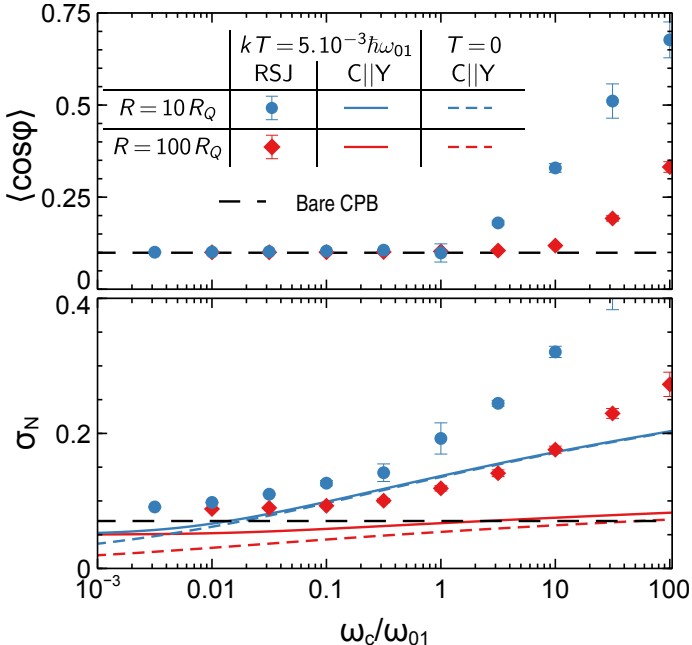

**Figure 3.** Dependence of the expectation values $\langle\cos\varphi\rangle$ (top panel) and $\sigma_N$ (bottom panel) with the cutoff frequency $\omega_c$ of the Ohmic bath, for an RSJ with $E_C = 10\,E_J$, $R = 10\,R_Q$ (blue) or $R = 100\,R_Q$ (red) at the low temperature $kT = 0.005\,\hbar\omega_{01}$. The black dashed lines correspond to the predicted values for the ground state of the bare CPB. In the bottom panel the colored lines show the predicted charge fluctuations $\sigma_{N,C||Y}$ in absence of junction conduction, both at the simulated temperature (solid lines) and $T = 0$ (dashed lines). One observes that at low cutoff the expectation values tend to those of the CPB independently of $\omega_c$ while at large cutoff the expectation values depend on $\omega_c$, with the superconducting character of the junction increasing with $\omega_c$.

Our method further allows to simply work out the dc linear response of the RSJ to a current bias. Indeed, adding a dc current source $I_b$ to the system adds a potential term $-\varphi_0 I_b\varphi$ to the Caldeira-Leggett Hamiltonian (1), and this term directly carries over to our path integrals, shifting the

minimum of the potential of the fictitious fluxonium away from $\varphi = 0$. With this term added, for small bias current $I_b \ll I_0 = E_J / \varphi_0$, stochastic Liouville numerics (see Fig. 4) yield $V \propto \langle N \rangle = 0$ and $I_0 \langle \sin \varphi \rangle = I_b$ up to numerical accuracy, corresponding to a supercurrent flow through the junction.

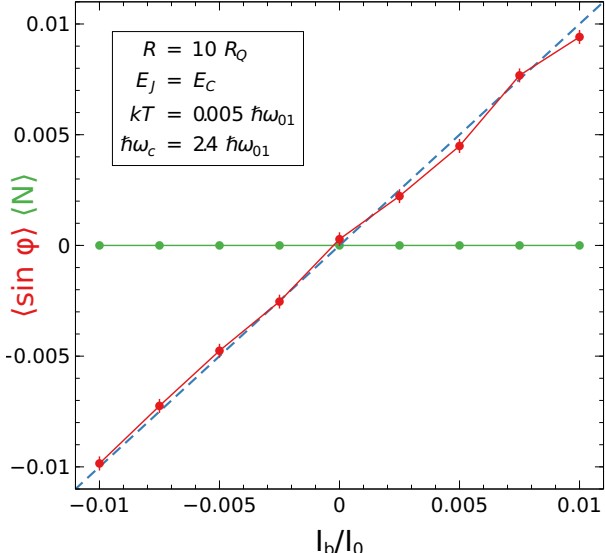

**Figure 4.** Example linear response expectations values of the current $I_0 \sin \varphi$ through the junction and the charge $2eN$ on the capacitor (proportional to the voltage across the junction) when adding a small bias current $I_b \ll I_0$ to the system. With a resistance $R = 10\,R_Q$, the junction would be in Schmid's "insulating phase", but our exact approach unambiguously yields a superconducting response.

# 5 Discussion

The superconducting linear response shown in Fig. 4 explains why RSJ experiments [17, 35] may observe signatures of a finite dc supercurrent that saturates at low temperatures for parameters that were previously believed to have an insulating ground state. As said above, a small current bias slightly shifts the global minimum of the potential of the fictitious fluxonium away from $\varphi = 0$, but the states remain localized in phase (this argument is critically examined in Appendix E.5), so that the dc voltage remains zero. Given Ohm's law for the resistance and current conservation, this implies all the bias current flows through the junction and that the junction hence has an infinite conductance. Instead of this junction conductance, many previous works considered that $\lim_{\omega \to 0} \omega \langle \varphi(\omega) \varphi(-\omega) \rangle$ (the so-called mobility) would give the zero-bias resistance of the junction in parallel with the resistor and found it saturating to $R$ when $R > R_Q$, thus seemingly indicating an insulating junction with zero conductance. Our results clearly contradict either these mobility results, or their interpretation. Indeed, we explain in D.1 why the mobility was incorrectly evaluated, and, in E.1, that the mobility is not a proper *observable* of the system, anyway. The static linear response of the RSJ can also be worked out using Kubo's standard framework [Altimiras, in preparation]; The latter approach, not invoking the action, tracing over the bath, etc, provides an alternate demonstration that the ground state linear response of the junction is superconducting for all parameters.

Our numerical results further show that the $R \to \infty$ limit of the RSJ smoothly recovers the well known physics of the CPB family of Josephson qubits, as expected for a vanishing perturbation. In addition, we observe that in the RSJ with a finite shunt resistance, at low temperatures, the effective Josephson coupling $E_J \langle \cos \varphi \rangle$ saturates to a value larger ($\geqslant$) than in the bare CPB and the rms charge fluctuations on the capacitor $\sigma_N$ saturate to values larger ($\geqslant$) than in the bare CPB, and strictly larger than in the $C \| Y$ circuit, for all the parameters we tested. This establishes that, in the Caldeira and Leggett model with an Ohmic environment having a finite UV cutoff frequency, the shunted Josephson junction's ground state is superconducting and actually *more superconducting* than the bare CPB junction.

Our results further show that the JJ's low-$T$ superconductivity increases at large cutoff of the Ohmic bath. The observed trend is actually simply explained by the counter-term localizing the phase more and more tightly (since $E_L \propto \omega_c$), which would ultimately yield a perfectly localized classical phase (and a superconducting junction) when $\omega_c = \infty$. The trend can also be equivalently explained by the logarithmic increase with $\omega_c$ of $\sigma_{N,C||Y}$ (Eq. (17)), the environment-induced charge fluctuations on the capacitor, which provides more charges states for the Josephson coupling mechanism, driving up both $\sigma_N$ and $\langle \cos\varphi \rangle$ (eventually reaching 1). Hence, we conclude a Markovian bath would yield a classical phase JJ with the maximal effective Josephson coupling $E_J$, for all values of the resistance. Although this result can be understood simply, it has surprisingly not been realized so far, to the best of our knowledge. Indeed, in the literature that predicted that transition, it is widely assumed without discussion that a strictly Ohmic bath with no UV cutoff would be appropriate for predicting the RSJ ground state (yet, not finding a fully localized phase, for reasons explained below). This prejudice that the bath's UV cutoff would be irrelevant is most likely due to assuming that the junction's capacitance by itself would sufficiently squash the high frequency fluctuations in the system. However, this is not the case since charge fluctuations on the capacitor diverge at infinite cutoff (see Eq. (17)).

Our findings can be globally explained qualitatively by arguing that connecting a resistor to a CPB can significantly affect the ground state of this nonlinear oscillator only if the environment impedance $Z(\omega) = Y^{-1}(\omega)$ is comparable to or lower than the effective impedance of the unshunted CPB at its plasma frequency, such that it can reduce the phase fluctuations across the junction. If furthermore the phase fluctuations of the bare CPB are initially large ($E_C \lll E_J$) the reduction of the phase fluctuations due to the resistor leads to an increase of $\langle \cos\varphi \rangle$, reducing the junction's effective inductance $L_J^{\text{eff}} = (\varphi_0^2)/E_J \langle \cos\varphi \rangle$, and hence its effective impedance, which in turn bootstraps the reduction of the phase fluctuations. Here, the method yields the exact self-consistent solution for these environment-modified fluctuations, similar to Ref. [36], but not restricted to Gaussian fluctuations only. The linear impedance of the bare CPB can be estimated using

$$\frac{Z_{\text{CPB}}}{R_Q} \sim \frac{1}{2\pi} \sqrt{\frac{\langle \varphi^2 \rangle}{\langle N^2 \rangle}}$$

which would be exact for the harmonic oscillator, or as

$$\frac{Z_{\text{CPB}}}{R_Q} \sim \frac{1}{R_Q} \sqrt{\frac{L_{\text{eff}}}{C}} = \frac{1}{2\pi} \sqrt{\frac{2 E_C}{E_J \langle \cos\varphi \rangle}}$$

both of which evolve from $\frac{1}{2\pi} \sqrt{\frac{2 E_C}{E_J}} < 1$ when $E_C \ll E_J$ to $\propto \frac{E_C}{E_J} \gg 1$ when $E_C \gg E_J$. This roughly explains at which resistance value the upturn of $\langle \cos\varphi \rangle$ occurs in Fig. 2. Yet, for $E_J/E_C \ll 1$, resistances much larger than the above estimates of the CPB linear impedance already induce a substantial change of $\sigma_N$ compared to the bare CPB, an effect dependent on the cutoff $\omega_c$ (See Fig. 3).

The temperature dependence of $\langle \cos\varphi \rangle$ is strikingly non-monotonous for large $E_C/E_J$ ratio (see Fig. 1). Starting from low temperatures, it first shows a plateau corresponding to the zero point fluctuations, followed by an increase with a local maximum around $kT/\hbar\omega_{01} = 0.1$, before reducing and finally vanishing at high temperatures. In the experimental results of Ref. [17], a similar non-monotonous variation of the junction's admittance was observed. We believe the remarkable similarity of these features between the experimental data and our numerical simulations constitute a cross consistency check of the experiment and of the present theoretical approach. Although this effect is easily explained by the resistor allowing charge fluctuations on the capacitor (see Appendix C), to the best of our knowledge, no other theoretical work on the RSJ predicts such behavior.

As fully expected from the qualitative argument on the existence of a localized ground state for all parameters given in Sec. 3, our numerical results show no sign whatsoever of Schmid's dissipative QPT in JJs. In particular, we observe no change of behavior at or near $R = R_Q$. As well, equilibrium observables related to transport do not follow power laws of the temperature in the critical region of the expected QPT, which would be the numerical signature of that QPT [18]. Schmid's QPT

does not occur in the RSJ. This conclusion brings theoretical support to that of Ref. [17], based on experimental observations.

Having invalidated Schmid's prediction, we still need to explain how the entire previous theoretical literature on that question could be mistaken. In Appendix D.1, we can pinpoint the reason for which other authors predicted the unphysical transition. As can be expected, the reason is rather subtle; it involves two unfulfilled implicit assumptions. The first assumption is already mentioned above, it is the widespread prejudice that the bath's UV cutoff would be irrelevant, leading to consider an infinite cutoff. The second implicit assumption is that this infinite cutoff limit would commute nicely with other limits, which it does not. All the works on that subject which considered baths without cutoff from the onset (notably, Schmid [3]) fell in the same trap of non-commuting limits. These subtle issues with limits explains why these authors predict an insulating state instead of the classical phase limit we find above for a strictly Ohmic bath; it also explains the theoretical inconsistencies evoked in Appendix A and why experiments may not follow their prediction. Our results reveal that, contrary to the past literature, it is essential to take into account the finiteness of resistor's UV cutoff (and the ensuing non-Markovian dynamics) for correctly predicting the RSJ ground state, whatever the theoretical approach; it is not just a matter of taste or convenience.

Finally, the present work provides for the first time a reliable way of predicting the equilibrium behavior of JJs in presence of arbitrary linear environments –even frequency-dependent ones–, provided the impedance is not too small. In addition this technique can, in principle, be extended to address the dynamical response and out-of equilibrium behavior of the JJ. In the opposite small shunting impedance regime, the approach should be doable in the dual picture, considering the coupling of the JJ charge with the impedance's fluctuating voltage.

# 6   Conclusions

Starting from the Caldeira-Leggett Hamiltonian, we write a formally exact path integral expression for the reduced density matrix of a Josephson junction shunted by a resistor. The bath's counter-term present in this expression makes it clear that a localized ground state always exists, and hence that the QPT to a delocalized ground state for $R > R_Q$, as predicted by Schmid, cannot exist.

This approach lends itself to a numerical implementation yielding the equilibrium reduced density matrix and the expectation values of observables of the RSJ, and notably the current in the linear response. This provides the first workable method to predict quantitatively the behavior of the RSJ in a wide range of parameters where predictions were previously impossible or incorrect. The method can be extended to frequency-dependent environment impedances, and, in principle, also to dynamical situations.

Our results

- fully support the conclusions of Murani *et al.* that a resistive shunt with $R > R_Q$ does not render a Josephson junction *insulating*. Actually, a shunt resistor can only make a junction *more superconducting* than it would be in its absence,

- recover the CPB physics when the shunt resistance is made very large, as expected for a vanishing perturbation,

- reveal an unforeseen dependence of the junction's superconducting properties with the resistor's UV cutoff, which must therefore be taken into account for making sensible predictions for the RSJ,

- explain how an issue of non-commuting limits associated with considering a resistor with infinite cutoff led many previous works to erroneously predict an insulating ground state in the RSJ for $R > R_Q$.

Together with the experimental results of Ref. [17], this work should close decades of misunderstandings around Schmid's prediction applied to JJs, which raised mysterious paradoxes and controversies. In particular, one should no longer claim that JJs are becoming *insulating* in high

impedance environments. Likewise, explanations for environment-related phenomena in JJ should no longer refer to Schmid's QPT in order to avoid confusion regarding the physics at play. Beyond the question of the Schmid transition, the rigorous theoretical basis on which this work present work further allows to clarify theoretical questions on the RSJ that have long been a matter of debate in the community, notably regarding the symmetries of the phase states and their degeneracies.

# 7  Acknowledgments

We are grateful to H. Grabert, J. Stockburger, C. Ciuti, L. Giacomelli, F. Borletto, R. Riwar, N. Roch, X. Waintal, I. Snyman, M. Houzet, O. Maillet, S. Latil, C. Gorini and our colleagues of the Quantronics group at CEA-Saclay for stimulating discussions, comments and helpful inputs at various stages of this work initiated 6 years ago. This work is supported in part by ANR project Triangle ANR-20-CE47-0011-02.

# Appendices

# A  On the interpretation of Schmid's QPT in Josephson junctions

What Schmid saw as remarkable in his work [3], was the *localization effect* in one well at large enough friction. Indeed, at low friction, a delocalized particle was seen as no surprise since one expects to recover Bloch states in the vanishing friction limit.

However, the way in which Schmid's analogy was received by physicists familiar with the Josephson junction had a totally reversed "surprise factor" : The predicted localized JJ phase was seen as run-of-the-mill since it is just like the classical description of the JJ which the beginner first learns (although in this classical description, dissipation is not *needed* to have a superconducting device). On the contrary, the delocalized phase in the weak damping limit, which was the vanilla situation for Schmid's original particle, was *interpreted* as an extraordinary situation in which the JJ would turn *insulating* under the action of the resistance, even when the corresponding friction force is vanishing. Indeed, according to what became the standard interpretation of Schmid's analogy, *a JJ could only be superconducting when it experienced strong damping of its phase*, even if this was in contradiction with the classical understanding of the device and with already abundant experimental observations of supercurrents in unshunted, undamped, junctions (by far the easiest to make and measure). As a corollary, the situation where Schmid wisely considered the effects of dissipation should be marginal, quite unexpectedly turned into a fascinating fantasy, even though it meant abandoning the mere notion of a perturbative effect.

Unsurprisingly, early experiments attempting to assess the existence of the predicted "insulating state" always observed that large junctions (i.e. with $E_J \gg E_C$) current biased through a very large resistor were contradicting the prediction by remaining superconducting [37, 38, 39], with apparently no one ever pointing that it had always been so and had never seemed a problem before. For making these observations compatible with the above interpretation of Schmid's analogy, the QPT prediction was patched, arguing that large undamped junctions were to be understood as being in a very long lived metastable superconducting state, with their "true insulating ground state" totally unreachable in practice [38, 9], while smaller junctions would actually reach their "insulating ground state".

But even with this "metastability fix" for large junctions, the standard interpretation of Schmid's prediction for JJ still had major consistency problems:

- First, Cooper-pair-box type of superconducting qubits with small unshunted junctions (i.e. the $R \to \infty$ limit of the RSJ) are found superconducting, contradicting both the prediction and the fix. This is obvious in the case where the junction is replaced by a squid: the observed temperature-independent flux tunability of such qubits implies that a dc supercurrent is circulating in the squid loop, so that neither of its junctions can actually be *insulating* (same argument as in Ref. [17]).

- Secondly, the standard interpretation of Schmid's prediction also has a theoretical continuity problem in the limit of large-area junctions, where the anharmonicity of the qubit vanishes : there, the ground state behavior of a RSJ must come to match that of the parallel RLC circuit (with a linear superconducting inductor) whose ground state is superconducting because of the inductor, whatever the value of the shunt resistor.

To the best of our knowledge, the theoretical literature on Schmid's transition in Josephson junction has not properly discussed how the predicted insulating phase could be reconciled with the basic continuity expectations in these simple limits.

Given that the behavior of RSJs was already qualitatively very well known [40, 41, 42] long before the time of Schmid's prediction, it is hardly explainable in retrospect that the awkward consequences of the standard interpretation of Schmid's analogy were not immediately pointed as inconsistent with the established knowledge on JJs. It is even more surprising that for nearly forty years the community remained in a situation where it believed the standard interpretation of Schmid's QPT in JJs was indisputable, either ignoring the problems mentioned above, or not caring to resolve theses oddities.

## B  Generation of discrete noise increments with required correlations

For the Lorentzian $\mathrm{Re} Y(\omega)$ Eq. (2) we assume, the integral in (9) converges for $0 < \tau < \hbar\beta$ and admits the analytical solutions

$$
\begin{aligned}
S_{II}(-i\,\tau) &= \frac{\hbar}{\pi R}\omega_c^2\left(\mathrm{Re}\left[e^{-\frac{2i\pi\tau}{\beta\hbar}}\Phi\left(e^{-\frac{2i\pi\tau}{\beta\hbar}},1,\frac{\beta\hbar\omega_c}{2\pi}+1\right)\right]+\frac{\pi}{\beta\hbar\omega_c}\right) \\
&= \sum_{n=0}^{\infty} S_n\cos(\omega_n\tau)
\end{aligned}
\tag{18}
$$

where $\Phi$ is the Lerch transcendent function, $\omega_n = n\frac{2\pi}{\beta\hbar}$, and

$$
S_n = \frac{\hbar}{R}\omega_c^2\,\frac{(2-\delta_{n0})}{\beta\hbar\omega_c+2\pi n}.
\tag{19}
$$

The expressions in (18) are even and $\hbar\beta$-periodic in $\tau$ (and, of course, symmetric about $\tau = \hbar\beta/2$). At $\tau \sim 0$ these expression have a mild divergence $S_{II}(-i\,\tau) \sim R^{-1}\omega_c^2\log|\tau|$ (See Fig. 5). Note that other forms of $\mathrm{Re} Y(\omega)$ with a sharper cutoff, such as e.g. $\mathrm{Re} Y(\omega) = \exp(-|\omega|/\omega_c)/R$, even yield a finite $S_{II}(0)$, i.e. a finite variance for $\xi(\tau)$. This should clear worries about possible consequences of the mild divergence of $S_{II}$ in the Lorentzian case, since one expects that the overall behavior of a system should not depend on the precise shape of such cutoff.

For satisfying Eq. (10), the random noise $\xi(\tau_n)$ can be naively drawn as the real numbers

$$
\xi(\tau_n) = \sum_{m=0}^{M-1} R_m\cos(\omega_m\tau_n+\theta_m)
\tag{20}
$$

where $\theta_0 = 0$, $R_0$ is a normally-distributed random number with zero mean and variance $S_0$, and the $\{R_{m>0}\}$ are taken fixed as $R_{m>0} = \sqrt{2 S_m}$, with the $\{\theta_{m>0}\}$ random and uniformly-distributed in $[0, 2\pi)$. Then, the $\{\xi(\tau_n)\}$ ensemble (Eq. (20)) is efficiently obtained as the real part of the fast Fourier transform (FFT) of $\{e^{i\theta_m}R_m\}$. With this construction, the correlators are

$$
\begin{aligned}
\langle\xi(\tau_n)\,\xi(\tau_m)\rangle &= \sum_{j,k=0}^{M-1}\langle R_k R_j\cos(\omega_k\tau_n+\theta_k)\cos(\omega_j\tau_m+\theta_j)\rangle \\
&= \sum_{j,k=0}^{M-1}\frac{1}{2}\{\langle R_k R_j\cos(\delta\tau\delta\omega\,(j\,m-k\,n)-\theta_k+\theta_j)\rangle \\
&\qquad\qquad + \langle R_k R_j\cos(\delta\tau\delta\omega\,(k\,n+j\,m)+\theta_k+\theta_j)\rangle\}
\end{aligned}
$$

$$= \langle R_0^2 \rangle \frac{1}{2} + \sum_{j=1}^{M-1} R_j^2 \frac{1}{2} (\cos(\delta\tau \delta\omega \, j \, (m-n))) + \langle R_0^2 \rangle \frac{1}{2} \cos(2\theta_0)$$

$$= \sum_{j=0}^{M-1} S_j \cos(j\,(m-n)\,\delta\tau\,\delta\omega)$$

$$= S_{II}\left(-i\,(\tau_n - \tau_m)\right) - \sum_{j=M}^{\infty} S_j \cos(j\,(m-n)\,\delta\tau\,\delta\omega) \tag{21}$$

which apparently almost fits the requirement (Eq. (10)).

The first problem with this naive algorithm is the logarithmic divergence of $S_{II}(-i\tau)$ at $\tau = 0$, for the Lorentzian $\mathrm{Re}Y(\omega)$ we consider. This is solved by taking, instead of $S_{II}(-i(\tau_n))$, the averaged $\overline{S_{II}}(-i(\tau_n)) = \delta\tau^{-1} \int_{\tau_n - \delta\tau/2}^{\tau_n + \delta\tau/2} S_{II}(-i\tau_n) \, d\tau$ over the time steps of our discretization, which removes the weak divergence. This amounts to filtering the correlation function by convolving it by a rectangular function, and hence to multiply the Fourier coefficients $S_n$ by a sinc

$$S_n \rightarrow \bar{S}_n = S_n \,\mathrm{sinc}\,\frac{\delta\tau}{2}\omega_n = S_n \,\mathrm{sinc}\,\frac{n\pi}{M},$$

and to define the $\{R_m\}$ from these $\{\bar{S}_m\}$.

Even with such filtering, a second problem remains : when taking the FFT, we only sum the $M$ first Fourier coefficients so that the correlator we obtain deviates from the ideal value, as apparent in Eq. (21). When $M$ is large enough, this deviation leaves a noticeable systematic error only for the same-time correlator

$$\langle \xi(\tau_n)\,\xi(\tau_n)\rangle \;=\; \overline{S_{II}}(0) - \Delta$$

where the error $\Delta$ is

$$\Delta = \sum_{j=M}^{\infty} \bar{S}_j = \frac{\hbar}{R}\omega_c^2 \, \frac{M\,\mathrm{Im}\left(\Phi\!\left(e^{-\frac{i\pi}{M}}, 1, M\right) - \Phi\!\left(e^{-\frac{i\pi}{M}}, 1, M + \frac{\beta\hbar\omega_c}{2\pi}\right)\right)}{\pi\beta\hbar\omega_c}.$$

Such Dirac delta-like error can be easily corrected by applying a shift to all the Fourier coefficients entering our FFT, except the zero-frequency one which provides the correct baseline

$$\bar{S}_j \rightarrow \tilde{S}_j = \bar{S}_j - (1 - \delta_{0j})\frac{\Delta}{M-1}, \qquad j = 0, \dots, M-1.$$

The $\{R_m\}$ are finally evaluated from the $\{\tilde{S}_m\}$ in place of the initial $\{S_m\}$. With these corrections made, we compare the numerical correlations to the expected $\overline{S_{II}}(-i(\tau))$ in Fig. 5

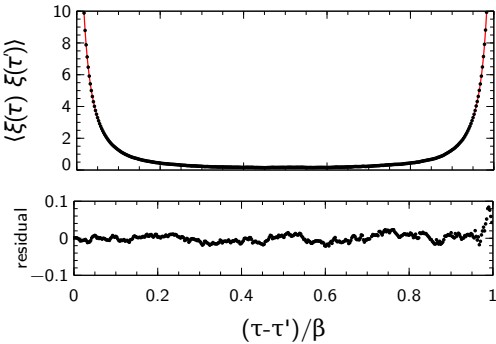

**Figure 5.** Comparison of the filtered (see text) theoretical current noise correlator in imaginary time for a Lorentzian $\mathrm{Re}Y(\omega)$ (red continuous line) and experimental correlator for $10^6$ drawings of a noise sequence (black dots). The bottom panel shows the difference between the numerical correlator and its expected value. Parameters are $R = 3\,R_Q$, $\beta\hbar\omega_c = 30$, 401 time steps.

# C Basis states, operator matrices and thermal expectation values for the bare CPB

In this appendix we evaluate thermal expectations values of some operators of the CPB, working in the eigenbasis. The CPB eigenstates can easily be obtained numerically in a truncated discrete charge basis, but below we rather obtain them analytically in terms of Mathieu functions [43, 44].

The Shrödinger differential equation for the bare CPB Hamiltonian in absence of offset charge is

$$E_C \, \Psi''(\varphi) - (E_J \cos\varphi) \, \Psi(\varphi) = E \, \Psi(\varphi). \tag{22}$$

This equation is a form of Mathieu's equation

$$f''(z) + (a - 2 \, q \cos 2 \, z) \, f(z) = 0,$$

whose solutions are known as special functions [45]. Furthermore, given that the potential is periodic in $\varphi$, Bloch's theorem implies the eigenfunctions of (22) are of the form

$$\Psi_{np}(\varphi) = \langle \varphi | n, p \rangle = e^{ip\varphi} \, u_{np}(\varphi),$$

where $n$ is a band index, and $p$ is the quasicharge (i.e. Bloch's quasimomentum), with $u_{np}(\varphi)$ a $2\pi$−periodic function of $\varphi$ (same period as the $\cos\varphi$ potential). If the CPB is not connected to anything, $p$ is fixed to zero, whereas when connected to a circuit that can let charge circulate, $p$ can fluctuate and take any value in $\mathbb{R}$.

Using knowledge from the solutions of Mathieu's equation, the eigenenergy $E_{np}$ of $\Psi_{np}(\varphi)$ is given by

$$E_{np} = \frac{E_C}{4} \, \lambda_{\chi(n,p)} \, (-2 \, E_J / E_C),$$

where $\lambda_\nu$ denotes the *Mathieu characteristic value* special function indexed by its *characteristic exponent* and

$$\chi(n, p) = n + n \bmod 2 + (-1)^n \, 2 |\text{frac}(p)|$$

is a function giving the characteristic exponents, such that the eigenenergies are sorted increasing with the band index $n \in \mathbb{N}$, and where the fractional value $\text{frac}(p) = p - \text{round}(p)$, with $\text{round}(p)$, rounding to the nearest integer. The $u_{np}$ functions themselves can be expressed as

$$u_{np}(\varphi) = \frac{e^{-ip\varphi}}{\sqrt{2\pi}} \left( \text{ce}_{\chi(n,p)}\left( \frac{\varphi}{2}, -2 \frac{E_J}{E_C} \right) + i \, (-1)^n \, \text{sign}(\text{frac}(p)) \, \text{se}_{\chi(n,p)}\left( \frac{\varphi}{2}, -2 \frac{E_J}{E_C} \right) \right),$$

where the Mathieu $\text{ce}_\nu$ and $\text{se}_\nu$ are respectively even and odd real functions of $\varphi$ [45]. Note that $\lambda_\nu$ has discontinuities when $\nu = \chi(n, p)$ is strictly an integer (*i.e.* when $2\,p$ is an integer), as well as either $\text{ce}_v$ or $\text{se}_\nu$; the eigensolutions to consider at these values in each band are then obtained as the limits when approaching the discontinuity. Our expressions with Mathieu special functions extend those of Ref. [43] to all quasicharges values, but differ from those of Ref. [44].

It is then straightforward to obtain the matrix elements of $N = -i \frac{\partial}{\partial \varphi}$ and $N^2$,

$$\langle n, p \, | \, N \, | \, n', p' \rangle = \delta \, (p - p') \left( \delta_{nn'} \, p - i \int_0^{2\pi} d\varphi \, u_{np}^*(\varphi) \frac{d \, u_{n'p}}{d\varphi}(\varphi) \right),$$

$$\langle n, p \, | \, N^2 \, | \, n', p' \rangle = \delta \, (p - p') \left( \delta_{nn'} \, p^2 - 2 \, i \, p \int_0^{2\pi} d\varphi \, u_{np}^*(\varphi) \frac{d \, u_{n'p}}{d\varphi}(\varphi) \right.$$

$$\left. - \int_0^{2\pi} d\varphi \, u_{np}^*(\varphi) \frac{d^2 \, u_{n'p}}{d\varphi^2}(\varphi) \right), \tag{23}$$

and those of any function $f(\varphi)$ are

$$\langle n, p \mid f(\varphi) \mid n', p' \rangle = \delta (p - p') \int_0^{2\pi} d\varphi \, f(\varphi) \, u_{np}^*(\varphi) \, u_{n'p}(\varphi). \tag{24}$$

Finally, we can evaluate thermal equilibrium expectation values from the thermal density matrix $\rho_\beta = e^{-\beta H} / \mathrm{Tr} e^{-\beta H}$ and the matrix elements of operators as

$$\langle A \rangle = \mathrm{Tr} \, \rho_\beta A = \sum_n \int_{-1/2}^{1/2} dp \quad e^{-\beta E_{np}} \langle n, p \mid A \mid n, p \rangle.$$

In qubits, the quasicharge charge $p$ has values externally imposed by the gate. The low impedance of the gate voltage is such that $p$ is nearly fixed and one should then only sum over the band index. On the other hand, if the qubit's "island" is not connected to a gate capacitance but rather to an element that can let charge circulate, $n_g$ can fluctuate and take any value. In Fig. 6, assuming either fixed charge offset or that $p$ (or $n_g$) can take any value, we plot the thermal expectation values $\sigma_N = \langle N^2 \rangle^{1/2}$ of the rms fluctuations of the charge $N$, and the Josephson coherence factor $\langle \cos \varphi \rangle$, which, being non-zero, are both indicators of the superconducting character of the unshunted CPB. One could also consider $\langle \sin^2 \varphi \rangle = (1 - \langle \cos 2\varphi \rangle)/2$, the fluctuations of the supercurrent, which is $1/2$ when the junction is insulating ($\varphi$ is delocalized, with all values equally probable) and smaller than $1/2$ when the junction has finite supercurrent fluctuations ($\langle \sin^2 \varphi \rangle = 0$ for the classical superconducting junction in absence of phase bias).

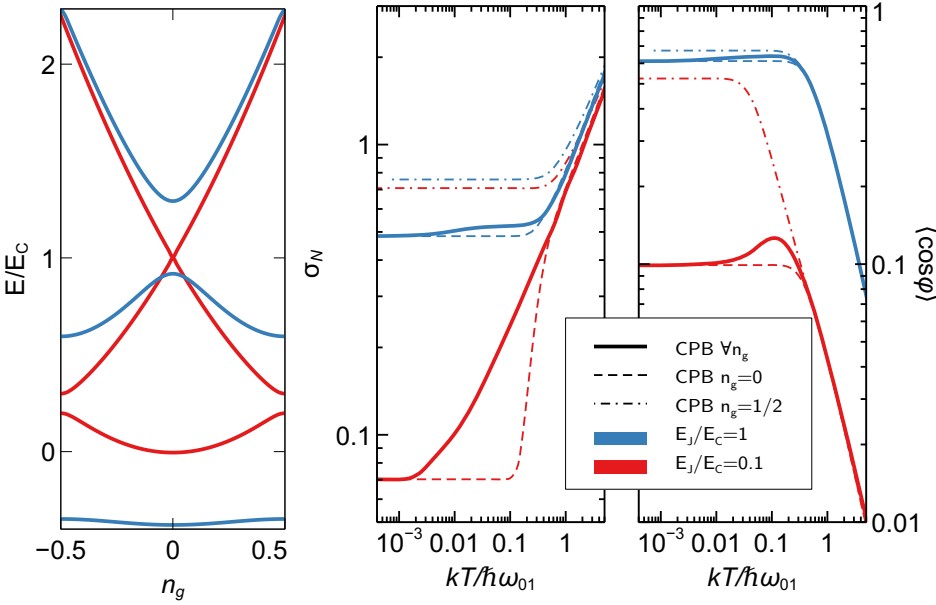

**Figure 6.** For all panels, red: $E_J = 0.1\,E_C$, blue: $E_J = E_C$ and $\hbar\omega_{01} = E_1(n_g = 0) - E_0(n_g = 0)$. Left panel: energy bands $E_0$, $E_1$ and $E_2$ (from bottom to top) of the CPB as a function of the gate charge. Middle (resp. right) panel, thermal expectation values of rms fluctuations of the charge $N$ (resp. Josephson coherence factor $\langle \cos \varphi \rangle$) as a function of temperature, for fixed gate charge $n_g = 0$ (thin dashed lines), $n_g = 1/2$ (thin dashed-dot lines), or (thick full lines) when allowing all gate charges.

This figure shows that at high temperatures, when $kT \gtrsim \hbar\omega_{01}$ (temperature larger than the separation of the first two bands at $n_g = 0$), the expectation values follow power laws $\sigma_N = \sqrt{kT/2\,E_C}$ and $\langle \cos \varphi \rangle \propto T^{-1}$, with values independent of whether $n_g$ is kept fixed or allowed to vary. In the opposite low temperature limit where $kT \ll E_0(n_g = 1/2) - E_0(n_g = 0)$ (the amplitude of the ground quasicharge band), expectation values saturate to a plateau corresponding to the zero point fluctuations of the ground state at $n_g = 0$.

In the intermediate temperature range, allowing charge fluctuations on the capacitor enhances both $\sigma_N$ and $\langle \cos \varphi \rangle$ with respect to the fixed $n_g = 0$ case, and this effect is most pronounced when $E_C / E_J$ is large (deep ground quasicharge band). For $\langle \cos \varphi \rangle$, this notably leads to a striking non-monotonous $T - $ dependence, with a local maximum at $k\,T \sim 0.1\,\hbar\,\omega_{01}$. This maximum is easily explained. In JJs with $E_C \gg E_J$, Cooper pair transfer occurs mostly through a second order tunneling process of quasiparticles, with a virtual intermediate state on higher charge parabolas. When allowing thermal fluctuations of the quasicharge away from 0 in the ground band, the energy difference between the ground and the lowest virtual excited state is reduced, hence increasing the effective Josephson coupling. At temperatures $k\,T \sim \hbar\,\omega_{01}$ or higher, the excited bands also get populated which then reduces the effective Josephson coupling.

# D  Comparing our effective action with that used in other works

The basis of our approach is the same as that used in most of the literature on Schmid's transition. Starting from the Caldeira-Leggett Hamiltonian (1) the dissipative bath is traced out using the Feynman-Vernon influence functional to obtain an effective action, from which one infers the properties of the system. However, several choices are possible, leading to different writings for the effective action in different works. In the first part of this appendix we show that we are describing the physics of the RSJ on the same grounds as in the rest of the literature on Schmid's transition, although with a more general kernel.

The total action for the system with the bath influence functional we consider, before performing the Hubbard-Stratonovich transformation, is (Eq. (5) and (6))

$$S_{\mathrm{Fl}}^E[\varphi] + \Phi[\varphi] = \int_0^{\hbar\beta} d\,\tau\,(\frac{\hbar^2}{4\,E_C}\,\dot\varphi^2 - E_J \cos\varphi + E_L\,\varphi^2)\ - \frac{1}{2} \int_0^{\hbar\beta} d\,\tau \int_0^{\hbar\beta} d\,\tau'\,\varphi(\tau)\,K\,(\tau - \tau')\,\varphi(\tau'), \tag{25}$$

with the kernel $K$ given by Eqs. (9) and (18). In the review Ref. [9], Schön and Zaikin write the effective action for the RSJ as

$$S[\varphi] = \int_0^{\hbar\beta} d\,\tau\,(\frac{\hbar^2}{4\,E_C}\,\dot\varphi^2 - E_J \cos\varphi) + \frac{1}{4} \int_0^{\hbar\beta} d\,\tau \int_0^{\hbar\beta} d\,\tau'\,K_\infty\,(\tau - \tau')\,(\varphi(\tau') - \varphi(\tau))^2 \tag{26}$$

(converting their notations to ours) where the first integral is the action of the Cooper pair box (with no counter-term) and where the kernel $K_\infty(\tau)$ has the form

$$K_\infty(\tau) = \frac{R_Q}{2\,R}\,\frac{\hbar}{\left( \hbar\,\beta \sin \frac{\pi\,\tau}{\hbar\,\beta} \right)^2} = \lim_{\omega_c \to \infty}\,K(\tau), \tag{27}$$

which corresponds to the particular case where the kernel Eq. (9) is evaluated with a purely Ohmic admittance $\mathrm{Re}Y(\omega) = 1/R$, without any UV cutoff (*i.e.* taking $\omega_c = \infty$ in (2)). The action used by Schmid [3] is the same as (26), with the influence kernel being furthermore the zero temperature limit of (27). For a moment, let us consider the influence functional of (26) with the more general kernel $K$ in place of $K_\infty$, and expand the square of phase difference. Then, using the facts that $K$ is even and periodic and that $\int_0^{\beta\hbar} d\,\tau\,K(\tau) = 2\,E_L$, one indeed formally recovers our form of the action with the counter-term,

$$\frac{1}{4} \int_0^{\beta\hbar} d\,\tau \int_0^{\beta\hbar} d\,\tau'\,K\,(\tau - \tau')\,(\varphi(\tau') - \varphi(\tau))^2\ =\ \int_0^{\beta\hbar} d\,\tau\,E_L\,\varphi(\tau)^2$$
$$- \frac{1}{2} \int_0^{\hbar\beta} d\,\tau \int_0^{\hbar\beta} d\,\tau'\,\varphi(\tau)\,K\,(\tau - \tau')\,\varphi(\tau'). \tag{28}$$

Thus, provided one uses the same finite-cutoff kernel (9) in our effective action (25) and in (26), the path integrals are equal; in particular, the ground states of these actions coincide. Hence, the localized ground state we find is also a valid ground state for Schmid's action with the finite-cutoff

kernel (9). This remains true when taking limits, e.g. the $T \to 0$ limit, or the $\omega_c \to \infty$ limit considered by other authors, where the ground state of our equations gets fully localized in phase (see Sec. 5).

## D.1 Why our results contradict previous theoretical work

Even though our equations appear compatible with those used by previous authors confirming Schmid's QPT prediction, these equations unambiguously lead us to conclude the opposite of these authors regarding that QPT. We discuss here more specifically the work of Werner and Troyer (WT) [14], who use the effective action (26) and apply the path integral quantum Monte Carlo numerical technique to investigate the predicted QPT in the RSJ. Hence, in principle, their technique and ours both calculate the same path integral, and a close examination should reveal why we come to different conclusions regarding the predicted QPT.

First, we observe that in spite of the formal equivalence of our two methods, it is not possible to directly recover and check WT's results with our stochastic Liouville method because, with the infinite cutoff kernel (27) they use, (i) $E_L = \frac{1}{2} \int_0^{\beta\hbar} d\tau\, K(\tau) = \infty$ and (ii) whatever the time discretization chosen, the strong $\tau^{-2}$ divergence of $K$ at short times prevents drawing small stochastic increments for a proper numerical integration of the Liouville equation. This unexpected non-equivalence of our two methods for the infinite cutoff limit considered by WT highlights that this limit requires careful handling (something one can hardly do by considering from the onset an infinite cutoff).

Precisely, in Sec. 4 and 5 of the main text, we investigate the role of the bath cutoff $\omega_c$ and show that, for reasons easily explained, the RSJ becomes more superconducting as $\omega_c$ increases, eventually reaching a classical phase state in the $\omega_c \to \infty$ limit. This trend and this limit we find are clearly the opposite of what would be needed to recover WT's result when $R > R_Q$, *viz.* a state with diverging phase fluctuations (see Fig. 3 in [46], the preprint version of [14]). However, we observe that these contradicting results are obtained by taking limits differently : in our approach, we first take the low $T$ limit of the path integral and then consider the infinite cutoff limit, while WT take the same limits in the reverse order. The different results we find indicate that these two limits do not commute.

In summary, by choosing to use the kernel (27) in the action (26), one implicitly assumes (i) an infinite cutoff would correctly describe an actual RSJ experiment, and also supposes (ii) the environment cutoff can be taken to infinity before evaluating the path integral and considering its low temperature limit. The results obtained with our exact approach reveal that neither of these implicit assumptions holds. These subtle unfulfilled assumptions suffice to explain why the phase delocalization QPT found by WT does not describe the actual physics of the RSJ. These unfulfilled assumptions similarly impact the results of all the other authors (Schmid, in particular) who consider the same $\omega_c = \infty$ limit from the onset.

In addition, we argue in the following Appendix, that the mobility evaluated by WT and other authors is not a proper observable and, therefore, it cannot be a valid "order parameter" for a QPT.

# E Symmetries and degeneracies in the RSJ

## E.1 Phase translation invariance

The Caldeira-Leggett Hamiltonian (1) is left invariant by the discrete translation symmetry that simultaneously shifts the junction's phase and all the bath oscillators' phases by the same multiple of $2\pi$:

$$\left. \begin{array}{rcl} \varphi & \to & \varphi + 2\pi k \\ \forall n, \quad \varphi_n & \to & \varphi_n + 2\pi k \end{array} \right\} k \in \mathbb{Z}$$

After tracing out the bath oscillators, a translational invariance with respect to the sole junction phase

$$\varphi \rightarrow \varphi + 2\pi k, \, k \in \mathbb{Z} \tag{29}$$

must remain for the states of the RSJ obtained from the RDM. Indeed, using (26) and (27) one can check this invariance is present in our path integral (4), before the Hubbard-Stratonovich transformation is made. However, the way the Hubbard-Stratonovich transformation is implemented, replacing a quadratic term in $\varphi$ by a linear term and leading to Eq. (11-12), breaks the above translation invariance. As a result, Eq. (11-12) implicitly select states that are centered at $\varphi = 0$. The $k \neq 0$ translated states would be obtained as solutions of the translated version of Eq. (11-12), where the potential terms in the fictitious system are translated. The effective action we use in the main text thus only yields a subset of the system states, and these localized states are infinitely degenerate through the $k$ values of this translation.

Although it may seem incorrect at first sight that our localized solutions do not have the translational symmetry builtin the original equations, this is neither an error nor a problem. Indeed, it is well known that when the eigenstates of a system are degenerate, some eigenstates may have a lower symmetry than the system as a whole (think *e.g.* of the $\ell > 0$ orbitals of the hydrogen atom, which do not have the spherical symmetry of the atom's Hamiltonian). A complete basis of degenerate subspace has the appropriate symmetries.

Let us consider the ground state RDM $\rho_0 = |\Psi_0\rangle\langle\Psi_0|$ obtained from Eq. (11-12), defining a reduced ground state $|\Psi_0\rangle$ localized and centered at $\varphi = 0$ (the diagonal $\langle\varphi\,|\,\rho_0\,|\,\varphi\rangle$ of this RDM is the square modulus of the reduced wave function $\Psi_0(\varphi) = \langle\varphi\,|\,\Psi_0\rangle$, i.e. the density of probability of the phase in this ground state). Then, following the above translation arguments, all the translated states $\left\{e^{-i2\pi k\hat{N}}|\Psi_0\rangle, k \in \mathbb{Z}\right\}$ are also valid reduced ground states of the RSJ. From the ensemble of these translated states, one can furthermore construct fully delocalized Bloch states with a dimensionless quasicharge $q$

$$|\Phi_q\rangle = \mathcal{N}\sum_k e^{-i2\pi k(q+\hat{N})}|\Psi_0\rangle \tag{30}$$

(with $\mathcal{N}$ the usual normalization factor of Bloch states) which have the same energy as $|\Psi_0\rangle$ for all values of $q$, and are hence also degenerate ground states. Thus, the RSJ has a flat quasicharge ground band, unlike the CPB (this difference is discussed more in detail in E.4). The fact that we can exhibit both localized and delocalized ground states illustrates the abstract argument given in Ref. [17], that any valid equilibrium state of the RSJ ought to be representable both as a localized or a delocalized state, based on formal arguments on unitary transformations developed earlier in Ref. [47, 48].

This localized|delocalized duality is further confirmed by the fact that one can come up with different (yet equivalent) equations for that system and which directly yield only delocalized states, for all parameters. Notably, it is possible to transform our effective action (25) ($\Leftrightarrow$ Eq. (5)-(6)) to a mathematically equivalent form :

$$S_{\mathrm{CPB}}^E[\varphi] + \tilde{\Phi}[\varphi] = \int_0^{\hbar\beta} d\tau \, (\frac{\hbar^2}{4\,E_C}\,\dot{\varphi}^2 - E_J\cos\varphi) \, -\frac{1}{2}\int_0^{\hbar\beta} d\tau \int_0^{\hbar\beta} d\tau' \, \varphi(\tau)\,k\,(\tau - \tau')\,\varphi(\tau'), \tag{31}$$

where the parabolic counter-term potential in the first term has been removed (*i.e.* it is the action of a CPB instead of a fluxonium) and absorbed in the modified influence functional $\tilde{\Phi}$ (See e.g. [30] or [29]). The kernel $k$ of this modified influence functional is related to the kernel $K$ (7)-(9) of the influence functional we use in the rest of this work through

$$k(\tau) = K(\tau) - 2E_L \text{Ш}_{\hbar\beta}(\tau),$$

where the $\mathrm{III}_{\hbar\beta}(\tau) = \sum_{n=-\infty}^{+\infty} \delta(\tau - n\hbar\beta)$ is the Dirac comb of period $\hbar\beta$, such that $\int_0^{\beta\hbar} d\tau\, k(\tau) = 0$. This alternate writing of the effective action does not have the confining potential that localizes our states, and that may seem artificial to some. After performing the Hubbard-Stratonovich transformation on this effective action, the fictitious system is a particle in a periodic potential submitted to a random noise which causes the particle to diffuse. In equilibrium, the diffusion current must vanish and the probability to find the particle is equal in all the wells. This is confirmed by running the stochastic Liouville method (see Sec. 3.2) with this counter-term-free modified action, as shown in Fig. 7. We observe that, indeed, without the counter-term, the results resemble the delocalized Bloch states described above, with expectations values of observables matching those of the simulations performed with the counter-term, within the error bars of the simulations. Were the number of wells taken into account in the numerics to be increased, the ground state obtained with this modified kernel would tend to the perfectly periodic phase state $|\Phi_{q=0}\rangle$ (also known as a compact phase state), as reasoned above.

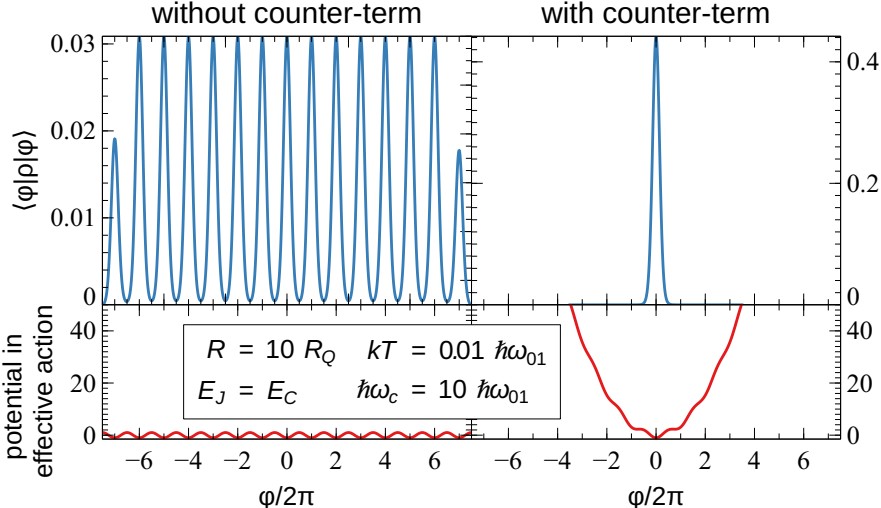

**Figure 7.** For the same set of parameters given in the label, we compare of the probability density $\langle \varphi\,|\,\rho\,|\,\varphi \rangle$ where $\rho$ is the RDM obtained from the stochastic Liouville method, computed with either (right panels) the effective action with the counter-term (25) used in this paper, or (left panels) the modified effective action (31) without the counter-term. The counter-term-free solution is close to a Bloch-like equal-weight superposition of repeated $2\pi$-shifted copies of the localized solution obtained with the counter-term, up to edge effects due to the limited number of cosine wells considered in the computation.

An important additional consequence of the translational invariance (29) of the effective action is that the *observables* of the system can only involve $2\pi$-periodic functions of $\varphi$. Thus, the phase itself, and all its moments, are *not* observables. In particular, the "mobility" $\lim_{\omega\to 0}\omega\,\langle\varphi(\omega)\,\varphi(-\omega)\rangle$ depends on whether one considers a localized or a delocalized ground state and its value hence does not describe the state of the RSJ in a meaningful way; contrary to what is generally considered in the literature, it cannot be an "order parameter" for Schmid's QPT.

## E.2 Charge translation invariance

A unitary transformation $U = \exp(i\varphi\sum_n N_n)$ applied to the Caldeira-Leggett Hamiltonian (1) yields the so-called charge gauge Hamiltonian $E_C(N - \sum_n N_n)^2 - E_J\cos\varphi + H_{\mathrm{bath}}$. The later Hamiltonian is invariant upon a translation of $N$ and any of the bath $N_n$ by the same arbitrary amount $\in\mathbb{R}$. Similarly to the phase translations, this means that, after tracing out the bath, for the reduced ground state $|\Psi\rangle$ obtained from the ground state RDM, any charge-translated copy $e^{iq\varphi}|\Psi\rangle$, $q\in\mathbb{R}$ is also a valid reduced ground state, with the same observables. This is another way to establish the degeneracy of the ground states with respect to the quasicharge, i.e. the flatness

of the quasicharge ground band. In other words, one can circulate an arbitrary charge in the loop formed by the junction and the resistance, without changing the properties of the system.

### E.3 In the RSJ, "the" ground state has no definite symmetry with respect to the phase.

The material in this Appendix shows that the infinite ground state degeneracy of the RSJ makes it pointless to debate about what ought to be "the" symmetry of "the" ground state with respect to the junction phase. This should also settle the longstanding debate on wether a galvanically-connected non-superconducting environment forces one to consider the JJ phase as an "extended" variable (as opposed to "compact" in the CPB). As explained above, the junction phase in the RSJ is not an observable; there is no actual physical meaning associated to the localized or delocalized symmetry of a given state in the phase representation, and, correspondingly, no measurement can assess whether it is localized or delocalized, extended or compact. It can also be regarded as the reason why Schmid's prediction of a localization|delocalization transition was erroneous: it does not make much sense. The translation invariance and the ensuing multiplicity of the ground states in the RSJ clearly differentiate this system from the akin spin-boson problem regarding the possibility of a spontaneous symmetry breaking caused by dissipation.

### E.4 Recovery of the CPB physics in the $R \to \infty$ limit

How is it possible that the CPB has a ground quasicharge band with a finite depth $E_S > 0$ (the so-called phase slip energy), while the RSJ has a flat ground band, even though in the large $R$ limit the effect of the resistance is perturbative and one expects to recover the CPB physics in the RSJ, i.e. with a finite $E_S$? The explanation is simple, actually: in the CPB, at non-integer $q$, the ground state has a finite charge on the capacitor (and there is a voltage across it); at this point, if a resistor is connected across the junction, such a finite static charge is no longer possible as the resistor drains it away in a few $RC$ time constant. Hence there is no contradiction: the RSJ eventually reaches its flat band ground state at times scales longer than $RC$, but when the resistance increases this takes longer and longer. In the $R \to \infty$ limit this relaxation to the flat band ground state would take an infinite time and one indeed recovers the CPB physics with a static gate charge. In other words, dc charging effects only occur in systems with an island; in the RSJ, a resistor with $R < \infty$ suppresses the CBP island.

Could it be that for $R_Q < R < \infty$, the symmetries of the Hamiltonian get spontaneously broken, such that the ground state is no longer invariant w.r.t. to charge translation, resulting in a non-flat quasicharge ground band? Then, as just discussed, there would be quasicharge states in this ground band having a static (dc) charge (and a voltage) on the capacitance, and that would imply that *both* the junction and the resistor became insulating. Thus, a transition making the ground band unflat cannot happen for $R < \infty$, since it would contradict the behavior assumed in the first place for the resistor.

### E.5 Stability of the superconducting ground state at finite current bias

In this subsection we critically examine the superconducting linear response of the RSJ presented in Sec. 4 and 5. The Hamiltonian of the current-biased RSJ is

$$ H = E_C N^2 - E_J \cos\varphi + \sum_n 4\,e^2\,\frac{N_n^2}{2\,C_n} + \varphi_0^2\,\frac{(\varphi_n - \varphi)^2}{2\,L_n} - \varphi_0 I_b \varphi. $$

At $I_b \neq 0$, the potential of this full Hamiltonian is not bounded from below. In that case, the localized states we obtain from our stochastic Liouville method, slightly off-centered from $\varphi = 0$, can at best only be metastable (just as in the standard tilted washboard image of current-biased JJs). However, in our approach, no runaway to a dissipative state can occur because the states are confined by the counter-term. How confident can we be that this superconducting linear response result we find is not merely an artifact of the method?

By applying a time-dependent unitary transformation $U(t) = e^{-iI_b\varphi t/2e}$ to the above full Hamiltonian, we obtain the transformed time-dependent Hamiltonian

$$\tilde{H} = U H U^\dagger + i\hbar \dot{U} U^\dagger = E_C \left( N + \frac{I_b}{2e} t \right)^2 - E_J \cos\varphi + \sum_n 4\,e^2 \frac{N_n^2}{2\,C_n} + \varphi_0^2 \frac{(\varphi_n - \varphi)^2}{2\,L_n}$$

where the potential is bounded as in (1) and where bias current now appears as a linear-in-time "offset charge" for the CPB. The above Bloch state $|\,\Phi_q\,\rangle$ (30) with the quasicharge $q = I_b t/2e$ is an exact (reduced) ground state for $\tilde{H}$ at time $t$. Hence, as long as it can follow adiabatically its flat quasicharge ground band, the system will remain in a zero-voltage state and sustain a (super)current flow. Such adiabaticity is guaranteed by general theorems [49, 50] at vanishing bias, but the bias current range actually allowing adiabatic dynamics cannot be determined in our equilibrium approach. The linear-response displaced localized states obtained with our stochastic Liouville method in Fig. 4 correspond to this adiabatic evolution; they are indeed valid metastable solutions, at least at vanishing bias. This confirms that the $I - V$ characteristic of the RSJ is vertical at the origin.

In their work, Caldeira and Leggett precisely considered these metastable states and explained quantitatively how dissipation reduced quantum tunneling of the phase [1], thereby increasing the lifetime of the metastable state and the ability of the junction to sustain a supercurrent. This is consistent with our conclusion in Sec. 5 that, at equilibrium, a junction shunted by a resistor is always more superconducting than an unshunted junction. Yet, when increasing the current bias $I_b$, at some point, adiabatic evolution in the ground band can no longer be maintained (in any case, one can expect a hard limit at $|I_b| \leqslant I_0$); the metastable localized ground state then occasionally experiences phase slips due to macroscopic quantum tunneling (at $T = 0$) through the cosine potential barrier and the supercurrent branch gradually or suddenly departs from zero volt at some finite current $\leqslant I_0$ dependent on the system parameters. At finite temperature thermal activation further causes phase slips and the overall behavior of the RSJ is in agreement with literature on the Josephson junction that largely predated Schmid's work [40, 41, 42].

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
