# Peer review of "Absence of a dissipative quantum phase transition in Josephson junctions: Theory"

_SciPost Physics_

## Round 2 · Referee Report · Roman-Pascal Riwar · 2024-4-4

Report
The authors revisit the phenomenology of the Caldeira-Leggett model applied to a resistively shunted Josephson junction, and calculate with numerical methods expectation values of the operators N^2 and cos(\phi) to track renormalisation effects, notably the fate of the renormalisation of the Josephson energy E_J. I think the work is clearly written, and mostly technically correct regarding the mathematical framework, and the numerically obtained results (some typos and subtleties will be commented on further below). Also, there are some potentially interesting aspects that are discussed with respect to the presence of cutoffs in the bath spectral density. But on the other hand, there are important points where I find the interpretation of those results either unclear or in some cases, with the utmost respect, problematic. Before I can make any further recommendation, I would therefore kindly ask the authors to address these points.
Specifically, the authors make the following key qualitative statements based on their results, which I try to summarise as accurately as possible like this:
1) In the limit of 1/R->0 (referred to as the perturbative limit by the authors), the system should return to the its completely uncoupled version — here, the fluxonium with bare parameters.
2) In the existing literature, this perturbative limit was not taken correctly, and the resulting phase diagram (in particular the insulator regime) are not correct when going towards 1/R-> 0.
3) They find that cutoffs in the Ohmic bath are important to resolve 2) (due to a non-commuting order of limits).
4) Cutoffs lead to an absence of an observable phase transition.
My assessment of these points in a nutshell: I completely agree with 1). As for 2), I partly disagree with the authors, even though I recognise that some of the existing literature was perhaps misleading on this aspect. Related to 2), I do not agree with statement 3). Statement 4) on the other hand seems scientifically sound, but I think the authors could flesh this point out more.
I now provide detailed arguments for my above assessment of points 1-4. This argumentation heavily relies on the book by Altland and Simons (“Condensed Matter Field Theory”, 2nd edition, Cambridge University Press), which I found particularly helpful and instructive in my own efforts to understand the topic of dissipative phase transitions. I will occasionally refer to explicit book pages in square brackets whenever I deem it necessary.
To start off, as indicated above, I of course agree with the authors fully and without reservations on point 1). In fact, this point is clear right from the start, when considering the action the authors give in their Eq. (4): for 1/R=0, we receive the action of the uncoupled system. Now, it is also true that the dissipative phase transition is very often depicted as E_J being either renormalised to 0 (insulating phase, R_Q/R<1) or to infinity (superconducting phase, R_Q/R>1) [see also page 425, Eq. (8.15) in Altland and Simons]. I fully concede that this seems to go against common sense. But I hope I can convince the authors with the following couple of lines of text, that these results to not refer to the bare energy scales, but only to energy scales with respect to other energy scales. In other words, I am convinced that one can revisit the standard literature treatment [see pages 422-425] with respect to bare energy scales, and resolve this issue without the need of adding cutoffs in the Ohmic bath.
In particular, it is illustrative to Fourier transform the quadratic part of the action (that is, everything except the Josephson energy). Thus, we find the action in frequency space as \int d\omega f(\omega) \phi(\omega)\phi(-\omega), where \omega are the Matsubara frequencies (which are of course strictly speaking discrete, but for small temperatures, when can go to the continuum limit). The function f(\omega) contains contributions from the charging energy, the inductive energy, and the dissipative kernel, which are all added up. The charging energy contributes with ~ \omega^2/E_C, where the \omega^2 comes from the time-derivatives in real-time. The inductive energy contributes a constant term ~ E_L. For a true Ohmic bath (i.e., without intrinsic cutoff), the dissipative interaction contributes a term linear in frequency, ~ R_Q/R*|\omega|. I postpone the discussion of a cutoff in the Ohmic bath to later.
Already at this stage, it is important to notice that the capacitive and inductive terms cure potential divergencies, either in the low-frequency infrared sector (where E_L is dominant), or in the high-frequency UV sector (where 1/E_C is dominant). The seemingly problematic result that the authors criticise comes precisely from discarding both the capacitive and inductive energies and only considering the dissipative contribution. This can be seen as follows. In the limit of small E_J, one can deal with the nontrivial cosine term in the form of a partial resummation of diagrams [see page 423, Eq. (8.14)]. Here, tracing out a certain frequency window in the partition function results in an effective renormalization of E_J of the form exp(-c)E_J, with c ~ \int d\omega 1/f(\omega). When now discarding all but the dissipative term, we get the well-known renormalisation where the ratio R/R_Q appears as the critical exponent, with precisely the seemingly counterintuitive/problematic behaviour for R_Q/R<1 (where E_J tends to zero). But again, I want to caution the authors, that this result has to be taken with a grain of salt, and stems from discarding the terms due to nonzero 1/E_C and E_L. Including those, we find first of all, that the integrand 1/f(\omega) for finite 1/R is always smaller than at 1/R=0. Moreover, due to finite 1/E_C and E_L, it does no longer diverge at 1/R=0. Consequently, the coefficient c is monotonically decreasing with increasing 1/R, its maximum (which is no longer infinite!) being at 1/R=0. Hence, for the full calculation, the procedure seems to suggest that the effective Josephson energy exp(-c)E_J is smallest (!) for 1/R=0 (in the absence of the bath). How can this be understood? Well, by realizing that the expression exp(-c) E_J is not the actual, physically measurable renormalization of E_J — instead, this is merely us performing the trace over all frequencies to compute the partition function. What is actually physically relevant is the fact that exp(-c) increases when switching on a finite 1/R. Consequently, in terms of pure energy scales (and not ratios of energies), the actual renormalised Josephson energy has to be computed by comparing the value for exp(-c) E_J at finite 1/R with respect to zero 1/R. That is, the actual relevant quantity is exp(-[c-c0]) E_J with c0 being the coefficient c evaluated at zero dissipation, 1/R=0. This function can only increase. And indeed, this is what the authors observe in their Fig. 2 (the panel showing the expectation value of cos(\phi)).
As an intermediate summary, we find that the perturbative limit is always guaranteed to be well-defined, even for a truly Ohmic bath without cutoff, due to the regulatory role played by the 1/E_C term in the UV frequency window [see also footnote 8 on page 422]. This is a first example where I agree with the mathematical result by the authors (their calculation of cos(\phi) yields exactly what I would expect), but not with the interpretation of why this result emerges.
Now, we come to the role of cutoffs in the bath spectral function. As outlined above, this cutoff is not actually necessary to make the limit 1/R=0 well-defined. But it may by all means play a role in the observation of a clean phase transition. Namely, by introducing an artifical cutoff in the spectral density of an otherwise Ohmic bath, the contribution of the bath inside the function f(\omega) changes qualitatively as follows. For frequencies below this cutoff, it still is linear, ~ R_Q/R|\omega|. But for frequencies above this cutoff, the dissipative contribution maxes out to a constant plateau value. This behaviour can be easily derived for a hard (theta function-like) cutoff. While I did not check this explicitly, I presume the Lorentzian cutoff the authors chose behaves similarly. At any rate, I expect that this maxing out changes the UV behaviour as follows: namely the 1/E_C term now becomes dominant for much lower Matsubara frequencies than for the pure Ohmic bath. Therefore, I believe that this reduces the renormalisation of E_J, not by altering the exponent (it still stays R/R_Q), but the ratio of frequencies that is exponentiated (which is now less extreme due to the lower UV cutoff). This smoothens the phase transition, such that it does not appear very sharp when evaluating observables. This is what I believe happens in Fig. 2, where we see a “shy” onset of an increase in E_J, but not a sharp transition.
However, this last finding is important, and here I see a potentially high merit in the work presented by the authors. Indeed a possible reason for the absence of an observable transition could be the smoothening out of the transition due to a deviation from a pure Ohmic bath. But in my point of view, this point is not clearly fleshed out and deserves a much more detailed analysis. Perhaps one could show the dissipative kernel in frequency space (Matsubara frequencies) and explicitly comparing it with the 1/E_C and E_L components, for the specifically chosen values. Also, I warmly recommend that the authors include a comparison of their numeric results for E_J<<E_C with semi-analytic results, which are easily accessible in that regime with the aforementioned partial resummation of the cos(\phi) term [see again page 423, Eq. (8.14)]. I think this could clear up this aspect, as the understanding of the renormalisation effect is very easily trackable in this regime (see above discussion of exp(-c)E_J). Also, (unless I overlooked something) did so far not see any estimate for an experimentally reasonable value for the Ohmic cutoff frequency (based on their experimental setups). If such an estimate exists, it might be good to provide it.
A further (related) aspect that I think could be addressed is the following. While it is theoretically by all means possible to calculate pure energies (and not ratios of energies) as the authors do in this work, I nonetheless wonder what the authors actually measure in their experiment. I understand that they calculate the inductive response of the junction, which yields the cos(\phi) expectation value, but surely the measured amplitude in front the cos(\phi) (the effectively observed Josephson energy) must be compared to another energy scale in their apparatus somewhere. Can the authors provide some information on that?
That being said, there remain a couple of more minor issues. The first is simply that the authors have an important typo in their Eq. (5): the first term should be proportional to 1/E_C and not E_C [very likely 1/(4E_C)].
The second point concerns the counter term ~ E_L. If the authors define the Hamiltonian as they do in Eq. (1), then the counter term must be absent in the action, Eq. (5). In fact, this counter term is “swallowed” by the kernel in Eq. (6), in the course of integrating out the bath degrees of freedom. This fact can be explicitly checked by doing the Gaussian integrals when starting from the full action. This fact is also explicitly discussed again in the book by Altland and Simons [see footnote 26 on page 130, and subsequent discussion, where the generic system-bath interaction has a similar quadratic counter term in the full action, which dissappears in the effective, traced-out action]. I consider this insofar a minor glitch as one can by all means add an actual physical inductive shunt (such that the bare system is not just the charge qubit, but the actual fluxonium). Hence, this does not change the behaviour at small 1/R. However, I wonder if this counter term has some influence on the phase diagram, as it seems to grow with increasing 1/R? Indeed, the authors at some point comment on the increasing E_L localising the phase. I fear that this is a spurious result, at least when accepting the Hamiltonian in Eq. (1). Finally, as a potentially interesting side note: indeed it seems that the partition function suffers from divergencies, if E_L is set exactly to zero. While there exist straightforward work-arounds for this problem (e.g., keeping E_L finite for the computation of the partition function, and only approach E_L->0 once the desired expectation value is computed), it might nonetheless be indicative of an invalid decompactification of the superconducting phase. But I suppose this subject is beyond scope for the present work.
Overall, I still have some reservations about the work presented by the authors. On the one hand, I appreciate the explicit statement that superconductivity is not literally suppressed for small R_Q/R. I myself am not 100% sure if this simple fact has sunken into the collective awareness of the community. But on the other hand, when reading pertinent standard literature on the topic (such as the here amply cited book by Altland and Simons), it is nonetheless implicitly clear that suppression of E_J is meaningful only in a specific RG context, and by no means to be taken literally. Moreover, as detailed above, I disagree with the authors on their analysis that the correct perturbative limit can only be attained by including a cutoff in the bath. For E_J<<E_C (when the cosine term can be treated quasi-perturbatively), this fact can be demonstrated analytically, as outlined above. Still, I find the inclusion of the cutoff interesting, as it indeed blurs out the phase transition, which might potentially be an experimental reality. But the authors likely have to improve the discussion of this aspect, to make it clear.
I therefore await a response by the authors, before making any definitive recommendation.

---

## Round 2 · Referee Report · Izak Snyman · 2024-4-24

Report
The authors calculate the ground state expectation value $\left<\cos \varphi\right>$ for a Josephson Junction (JJ) with an Ohmic shunt of resistance $R$. They find that $\left<\cos \varphi\right>$ is positive (and never zero) for all $E_J>0$. They assume that this is inconsistent with known results for the Schmid transition, namely that the order parameter $\lim_{\omega \to 0} \omega \left<\varphi(\omega) \varphi(-\omega)\right>$ is zero for $R < R_Q$ and finite for $R > R_Q$. They propose that the inconsistency is due to what they claim to be the previously unnoticed non-commutativity of taking the $T \to 0$ limit and imposing a finite UV cut-off.
The authors display the courage of their convictions, and do not shy away from acknowledging that their results contradict 40 years of theoretical literature.
I thank the authors for being so forthright. Currently, there certainly is confusion about the absence or presence of a dissipative quantum phase transition in the resistively shunted Josephson Junction (RSJ), although I am not sure that this confusion dates from 40 years ago. To clear this up, it helps enormously if people explain themselves straightforwardly, as the authors do.
I believe that the numerical results in the manuscript are valid, and that $\left<\cos \varphi\right>$ is indeed non-zero, for all $R$. However, with the utmost respect to the authors, I believe that the significance they attach to this result, including the claim that it proves existing studies of the Schmid transition wrong, to be off the mark. I will attempt to be as forthright as the authors, in explaining how I arrived at this conclusion. If I've made a mistake, this will allow the authors to pin-point exactly where I got it wrong.
- The authors find that $\left<\cos \varphi\right>$ is non-zero even for large shunting resistance. Is this result correct?
Yes I believe it is. The following elementary argument backs it up:
Let $H_0$ be the Hamiltonian (1) of the manuscript, with $E_J=0$. Let $\left|GS,0\right>$, and $E_{GS,0}$ be the ground state and ground state energy of $H_0$. Let $E_{GS}$ be the ground state energy of the full Hamiltonian $H$, and let $|GS>$ be the ground state of the full $H$.
From the variational principle, we have that $\left<GS|H_0|GS\right> > E_{GS,0}$.
Because the spectrum of $\cos \varphi$ ranges from -1 to 1, we expect that $E_{GS}$ is less than $E_{GS,0}$ by a finite amount, at least when $E_J$ is sufficiently large.
We can in fact prove this for all $E_J>0$:
$H_0$ commutes with the total charge $P=N + \sum_n N_n$ and therefore $\left|GS, 0\right>$ can be taken as an eigenstate of $P$. Thus $\left<GS, 0| \cos \varphi |GS, 0\right>=- i \left<GS,0| [\sin \varphi, P] |GS, 0\right> = 0$. According to the variational principle $E_{GS} < \left<GS,0| H |GS,0\right> = \left<GS,0| H_0 |GS,0\right> - E_J \left<GS, 0| \cos \varphi |GS, 0\right> = \left<GS,0| H_0 |GS,0\right> = E_{GS,0}$
Consequently, we have $\left<GS|H_0|GS\right> > E_{GS_0} > E_{GS}$ which implies $0 < \left<GS| H_0 - E_{GS} |GS\right> = \left<GS| H_0 - H |GS\right> = E_J \left<GS | \cos \varphi |GS \right>$, i.e. $\left<GS| \cos \varphi |GS\right> >0$.
- Has 40 years of literature gotten this elementary fact wrong?
No, as far as I can tell, only the deeply flawed Ref. 21, has ever claimed that $\left<GS| \cos \varphi | GS\right>$ is zero for certain large values of the shunting resistance (and large $E_C/E_J$).
-So what does the literature of the Schmid transition say about the Hamiltonian (1)?
The Hamiltonian (1) has a discrete translation symmetry. If one translates all the phases by $2 \pi$, i.e. $\varphi \to \varphi + 2 \pi$ and $\varphi_n \to \varphi_n + 2 \pi$, then the Hamiltonian is left invariant. When the coupling to the bath exceeds a critical value, i.e. when $R<R_Q$, this symmetry is spontaneously broken. The order parameter that is usually calculated is $\lim_{\omega \to 0} \omega \left<\varphi(\omega) \varphi(-\omega)\right>$ which is zero for $R < R_Q$ and finite for $R > R_Q$.
The intuitive physical picture is as follows. At large $R$ (weak coupling to the bath), the phase $\varphi$ is delocalized due to the unbroken discrete translational symmetry. At stronger coupling to the bath (smaller $R$), the bath ``measures'' the phase, and it localizes.
-But how can both sides be right. The authors say that their results contradict 40 years of theory?
$\left<\cos \varphi\right>$ is not expected to be a good order parameter. It can only detect breaking of full translation symmetry $\varphi \to \varphi + d\varphi$, not discrete translation symmetry $\varphi \to \varphi + 2 \pi$.
A few years ago, I studied the Hamiltonian (1) variationally. I was mislead by Ref [21] into thinking that $\left<\cos \varphi\right>$ is a valid order parameter for the Schmid transition, and was therefore intrigued when I found $\left<\cos \varphi\right> > 0$ at large $R$. I discussed this with a colleague, claiming that perhaps there is no transition in an RSJ at $R=R_Q$. However, shortly after the discussion, I had an insight that caused me to retract my claim regarding the absence of the Schmid transition. I therefore wrote my colleague an email explaining as follows: "I now doubt the correctness of my claim that only the superconducting phase can be characterized by non-zero <cos phi> and hence negative energy implies superconductivity. [...] The point is that in general, well-defined quasi-charge still leads to non-zero <cos phi>. Quasi-charge tells you that there is discrete translation symmetry between unit cells, whereas <cos phi> tells you what goes on inside each unit cell."
To explain what I wrote: quasi-charge here refers to the discrete translation operator $\exp[2 \pi i (N + \sum_n N_n)]$ which translates all phases by $2 \pi$. Well-defined quasi-charge means unbroken discrete translation symmetry, and a delocalized phase phi. The point is that a quasi-charge (or Bloch) eigenstate with discrete translation symmetry, still generally gives $\left<\cos \varphi\right>$ non-zero. This happens when $\varphi$ is more likely to be found at integer multiples of $2 \pi$ than elsewhere.
My colleague wrote back:
"it indeed looks obvious now as you said it, but it did not come to my mind during our discussion last week. Does it mean that the same object can be an insulator when probed by an external current source, but a superconductor when included in a loop and probed by an external flux ."
I believe this remark hits the nail on the head, and resolves the apparent paradox that the authors highlighted. The order parameter $\lim_{\omega \to 0} \omega \left<\varphi(\omega) \varphi(-\omega)\right>$, that is usually calculated, tells one whether the phase phi starts running if the RSJ is current biased by adding a term $I_\text{bias} \varphi$ to the Hamiltonian. If the phase runs, it means a DC voltage appears in response to $I_\text{bias}$. (As the authors probably know, the order parameter is in fact the associated DC resistance in response to a current bias.) The existing literature on the Schmid transition shows that at zero temperature, this resistance in response to a current bias drops to zero abruptly when the shunting resistance drops below $R_Q$.
As my colleague pointed out, this does not contradict the following: the Schmid transition is not revealed when instead of a current bias, a phase bias is applied instead. For instance, in a resistively shunted SQUID, that is threaded by a flux, a super-current is expected, regardless of the value of the shunting resistance. Why?
Consider the case where the discrete translation symmetry is not broken. When the flux through the squid is zero, the phase is more likely than not to be close to a multiple of $2 \pi$, and $\left<\cos \varphi\right>$ is non-zero. With a non-zero flux through the SQUID, it introduces an offset, while retaining a $2 \pi$ periodicity. As a result, the super-current $\left<\sin \varphi\right>$ is non-zero for a phase-biased junction, even though $\varphi$ is not localized.
-What about the $R \to \infty$ limit? The authors claim that the existing literature gets this wrong.
No, the authors' expectation about the $R \to \infty$ limit is consistent with the existing literature. To see this, note that the authors assume $\varphi$ is extended at the outset: In the Hamiltonian (1), the phase $\varphi$ must be extended, and cannot be compact, since the Hamiltonian is not periodic in $\varphi \to \varphi + 2 \pi$. The authors' CPB is a JJ with an infinitesimal galvanic coupling to a harmonic environment. So, for our purposes, there is no need to go into the subtle issue of when it is correct to view phi as extended.
Given all this, the ground state of the $R \to \infty$ Hamiltionian, $E_C N^2 - E_J \cos \varphi$, is a Bloch state, as the authors point out in their Appendix C, and $\varphi$ is delocalized as Schmid would have considered to be unsurprising. Yet $\left<\cos \varphi\right>$ is non-zero, as the authors point out in the main text. This is an elementary illustration of the remark my colleague made.
-In Appendix D the authors explain ``why our predictions contradict previous theoretical work''. Is their explanation correct or mistaken?
It is mistaken. The appendix focusses on the work in Ref. [14] by Werner and Troyer (WT). The authors assert that their own work addresses the same question as WT, but that they get a different answer from WT. However, WT calculate $\omega \left<\varphi(\omega) \varphi(-\omega)\right>$ while the authors calculate $\left<\cos \varphi\right>$. As explained above, it is no paradox that WT detect the Schmid transition - they calculate the correct order parameter to detect breaking of discrete translation symmetry - while the manuscript under review does not detect the Schmid transition: $\left<\cos \varphi\right>$ cannot detect breaking of discrete translation symmetry.
To substantiate the authors' contentions, the same quantity should be calculated with both methods. In this regard, I note the following.
I believe WT would have obtained the same result as the authors, had they calculated $\left<\cos \varphi\right>$. I base this statement on the fact that my argument above for finding $\left<\cos \varphi\right> > 0$ is quite general. It does not require a bath with a UV cut-off. Furthermore, if the phase trajectories that WT show in their Figure 1 are typical, it is clear that $\varphi$ spends the majority of its time close to multiples of $2 \pi$, and $\left<\cos \varphi\right>$ will therefore be larger than zero.
I further note that it is likely not possible to calculate $\omega \left<\varphi(\omega) \varphi(-\omega)\right>$ accurately with the authors' method. My reason for saying this is as follows:
The action in (4) has discrete translation symmetry under $\varphi \to \varphi + 2 \pi$ as it should, thanks to $\int_0^{\hbar \beta} d\tau K(\tau) = 2 E_L$. However, the action $\int_0^{\hbar \beta} d\tau (H_{FL} + \xi \varphi_0 \varphi)$ in equation (10) badly breaks this symmetry, due both to the term $E_L \varphi^2$ in $H_{FL}$, and the term proportional to $\xi$. It is only after averaging over $\xi$ is performed, that the symmetry is restored. The finite sampling over $\xi$ that the authors do, will not perfectly restore discrete translation symmetry, and is therefore probably not well-suited for studying a phase transition associated with this symmetry. At best, one might see $\left<\varphi^2\right>$ rapidly change from a large to a small value as $R$ is reduced below $R_Q$, provided $E_L$ is not too large.
The authors make the general claim that it matters whether a UV cut-off is present at the outset, or introduced after the $T\to0$ limit is taken. On general grounds, I do not think this is plausible: it would invalidate many well-established results obtained by Renormalization Group methods (Kondo a la Yuval and Anderson, BKT, etc.), and turn our field on its head.
More specifically, the authors claim that WT get incorrect results because they use a kernel $K(\tau)$ that diverges like $1/\tau^2$ at $\tau \to 0$, whereas the authors get correct results because their kernel is a regularized version, which only diverges like $\text{log} \tau$, but coincides with WT's kernel at times sufficiently sufficiently further away from 0 or $\hbar \beta$ than $1/\omega_c$. Firstly, this is pure speculation, since the same quantity was never calculated both ways. Secondly, the speculation is very likely incorrect: Apart from the arguments already given, I would like to point out that WT's method discretizes time on a lattice, and the lattice constant plays a similar role to $1/\omega_c$ in the manuscript under review. Furthermore, there is nothing unphysical about the $1/\tau^2$ divergence in the continuum limit, it simply imposes continuity of $\varphi(\tau)$, including between start and end points, since bosonic fields are periodic.
I further note that WT do include a capacitive term. I fully concur with the first referee that this provides a physical UV regularization: a shunting capacitor shields the JJ from high frequency electromagnetic fluctuations. So even if it matters whether one sends T to zero before or after introducing a high energy cut-off (which I dispute), WT's model has a physical UV cut-off from the outset.
-Is my report consistent with that of Riwar (the first referee)?
I believe it is. The bulk of Riwar's report discusses what one will observe if one measures $\left<\cos \varphi\right>$, and how to reconcile this with the fact that RG schemes such as discussed in Altland and Simon obtain a running coupling $E_J$ that goes to zero for $R>R_Q$ as lower and lower energies are traced out. I agree with the offered resolution in terms of "pure" vs. "relative" energy scales. I believe Riwar's statement that details about the UV cut-off "smoothens the phase transition, such that it does not appear very sharp when evaluating observables" refers to measuring $\left<\cos \varphi\right>$. I do not think it is intended to apply to the zero-temperature order parameter $\omega \left<\varphi(\omega) \varphi(-\omega)\right>$.
One point where I disagree with Riwar is regarding the counter-term $E_L\varphi^2$. According to Riwar, if the Hamiltonian is as in equation (1), the counter-term should be absent from equation (5). However, unless I am missing something, the counter terms in equation (5) is correct. An easy way to see this, is to note that without it, the action would not be invariant under $\varphi \to \varphi + 2 \pi$, as it should be, after integrating out $\varphi_n$.
-Where to now?
While the authors did some worthwhile numerical work, the central thesis of the work is wrong, and the manuscript cannot be accepted.
To refute my conclusion, the authors would at the very least have to implement Werner and Troyer's method, and use it to calculate $\left<\cos \varphi\right>$. If they can show that a careful calculation yields $\left<\cos \varphi\right>=0$, it would provide some justification for their thesis.
Failing the above, I could imagine a different manuscript, detailing the same numerical calculation, being acceptable for publication. This manuscript would present the authors' numerical method as a new tool to study the RSJ. It would then use this method can calculate <cos phi>. It would point out that $\left<\cos \varphi\right> > 0$ for $R>R_Q$, which naively is surprising, but in fact expected. It would then explore how to reconcile this with the existing literature, along the lines of the arguments presented by Riwar and myself, without claiming that the existing literature is wrong.
Recommendation
Ask for major revision

---

## Editorial Decision

unknown